# Suppress and Diversify: Refining Robust Pathways for Corruption Robustness

Jiangang Yang [* 1]  Wenhui Shi [* 1 2]  Xiaoran Xu [2]  Wenyue Chong [2]  Luqing Luo [1]  Jing Xing [1]  Jian Liu [1]

## Abstract

Model robustness against natural image corruptions is essential for safety-critical applications. While existing methods primarily focus on implicit representation learning, we provide the first systematic exploration of computational pathways to explicitly characterize internal robustness. We identify a progressive decay of robust features across network layers and establish a functional dependency between the prevalence of these features and model performance. To exploit these insights, we propose Suppress and Diversify (S&D), a non-intrusive refinement approach that enhances robustness by dynamically selecting robust pathways and diversifying them through symmetry-preserving transformations. S&D is architecture-agnostic, parameter-free, and incurs zero test-time overhead. Extensive evaluations across eight benchmarks demonstrate that S&D consistently improves performance across multiple vision tasks, diverse backbones, and complex real-world scenarios, highlighting its broad efficacy and scalability.

## 1. Introduction

Recent years have witnessed the success of neural networks in various computer vision tasks. However, recent works have revealed that neural networks are vulnerable to image corruptions (Hendrycks & Dietterich, 2019): image changes (e.g., noise, blur, or low lighting) that degrade visual quality can lead neural networks to make errors such as misclassifying a bicycle as a motorbike. Such vulnerability inevitably transfers to downstream 2D and 3D perception tasks (Michaelis et al., 2019; Kamann & Rother, 2021; Dong et al., 2023; Zeng et al., 2024), raising serious security concerns in safety-critical applications.

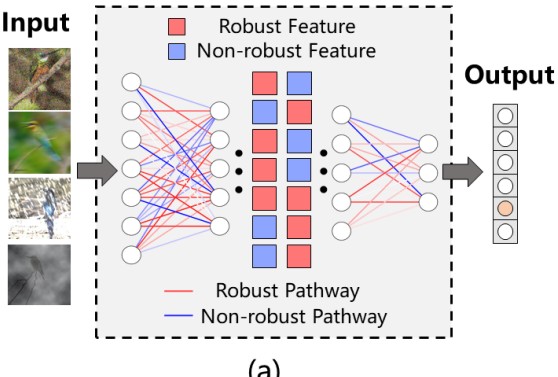

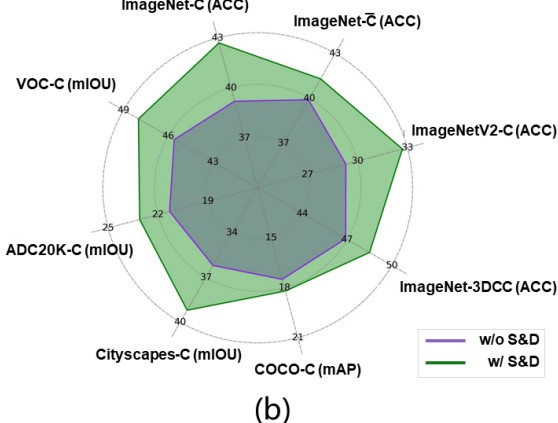

*Figure 1.* (a) The phenomenon of robust (non-robust) features for image corruptions . (b) The performance gains of the proposed approach across various computer vision tasks.

To solve this problem, prior approaches primarily focus on implicitly learning robust representations through data augmentation (Hendrycks et al., 2019; Modas et al., 2022; Qin et al., 2022), model regularization (Zhang et al., 2018; Guo et al., 2023), ensemble learning (Saikia et al., 2021; Diffenderfer et al., 2021), and biological principles (Teti et al., 2022; Dapello et al., 2020). However, explicit modeling of robust features under image corruptions remains largely under-explored. While adversarial robustness research has extensively studied feature disentanglement (Ilyas et al., 2019; Tao et al., 2022; Kim et al., 2023; Wang et al., 2024), these findings may not directly apply to natural corruptions, which exhibit global structural patterns and semantic shifts

[1]Institute of Microelectronics, Chinese Academy of Sciences, Beijing, China [2]University of Chinese Academy of Science, Beijing, China. Correspondence to: Jiangang Yang <yangjiangang@ime.ac.cn>, Jian Liu <liujian@ime.ac.cn>.

rather than local pixel-level noise. The nature of features under such structured distortions remains poorly understood, raising several critical questions: (a) *Do neural networks similarly exhibit distinct non-robust features under image corruptions?* (b) *Can we explicitly manipulate robust (non-robust) features to enhance corruption robustness?*

To address these questions, we provide a systematic analysis to verify the existence and impact of non-robust features. We first formalize the robustness of computational pathways via the local Lipschitz constant (Cisse et al., 2017; Fazlyab et al., 2019), revealing that robust and non-robust features are prevalent across the network. Our investigation identifies a progressive decay of robust features along the network depth, suggesting a non-homogeneous evolution of internal robustness. Furthermore, through linear probing and loss landscape analysis, we establish a functional dependency between robust feature prevalence and overall performance. These findings demonstrate that the aggregation of robust features dictates the robustness of downstream sub-networks and the full model under corruptions.

Building on these insights, we propose Suppress and Diversify (S&D), a non-intrusive refinement approach. The core principles are: (1) selecting pathways that generate robust features, and (2) diversifying them. Specifically, we introduce a Memory-aware Dynamic Selection Mechanism (MDSM) that maintains pathway groups in a memory bank, dynamically updated by a fitness score balancing feature invariance and discriminability. To prevent overfitting, a Structure-consistent Path Tweaking Strategy (SPTS) subsequently filters these pathways and generates augmented counterparts via symmetry-preserving transformations. This strategy enhances the diversity of robust representations while adhering to structural invariants. Finally, S&D ensures the network is guided by a diverse set of robust computational pathways, significantly improving performance under corruptions with zero test-time overhead.

S&D is a lightweight refinement requiring no inference-time overhead or structural modifications. We validate its efficacy across diverse benchmarks for classification, detection, and segmentation. S&D consistently enhances the intrinsic robustness of various architectures and provides orthogonal gains when integrated with state-of-the-art methods. Our contributions are summarized as follows:

- We provide the first systematic exploration of non-robust features under image corruptions, revealing how the prevalence of these features determines the robustness across sub-networks and the overall model.

- We introduce S&D, a novel training-time refinement that suppresses non-robust pathways and diversifies robust ones. S&D is architecture-agnostic and parameter-free, enabling plug-and-play integration.

- Extensive evaluations across diverse benchmarks and vision tasks demonstrate that S&D consistently enhances corruption robustness and OOD generalization.

## 2. Related Work

### 2.1. Understanding Corruption Robustness

There has been growing interest in understanding corruption robustness. Early work compares human and neural-network robustness under image corruptions (Geirhos et al., 2018b). A major line of research explains robustness through model bias, especially texture/shape bias: texture bias plays a key role in corruption performance (Geirhos et al., 2018a). Complementary perspectives analyze robustness in the Fourier domain by separating low- and high-frequency effects (Yin et al., 2019), and connect robustness with decision-boundary thickness (Yang et al., 2020). Large-scale evaluations show that several bias-based assumptions do not hold for network robustness (Gavrikov & Keuper, 2024). Closest to our focus, recent work implicitly models patch-induced non-robust features in Vision Transformers and shows that models exploit these patterns (Qin et al., 2022; Guo et al., 2023). However, no prior work explicitly models robust or non-robust features as a direct lens for understanding corruption robustness.

### 2.2. Enhancing Corruption Robustness

Enhancing corruption robustness is a major research focus, with benchmarks spanning multiple vision tasks (Hendrycks & Dieterich, 2019; Michaelis et al., 2019; Kamann & Rother, 2021). A dominant direction is data augmentation, which consistently improves robustness (Geirhos et al., 2018a; Yin et al., 2019; Guo et al., 2023). Aug-Mix (Hendrycks et al., 2019) mixes diverse image operations, and later methods extend mixing to style (Hendrycks et al., 2021; Zhou et al., 2024), spatial (Modas et al., 2022), and frequency domains (Yucel et al., 2023; Vaish et al., 2024). Other lines combine augmentations via ensembles (Saikia et al., 2021; Diffenderfer et al., 2021) or propose training recipes coupling augmentation with regularization (Wightman et al., 2021; Touvron et al., 2021). Beyond training pipelines, bio-inspired architectures and learning paradigms are explored (Dapello et al., 2020; Teti et al., 2022; Li et al., 2019; Safarani et al., 2021). Fine-tuning strategies also target corruption-robust sub-networks (Lee et al., 2022; Guo et al., 2022); among them, EWS (Guo et al., 2022), closely related to our approach, uses knowledge distillation with a model-specific controller to refine non-robust sub-networks. In contrast, our method requires no additional model structures and can be seamlessly integrated into existing deep-learning frameworks.

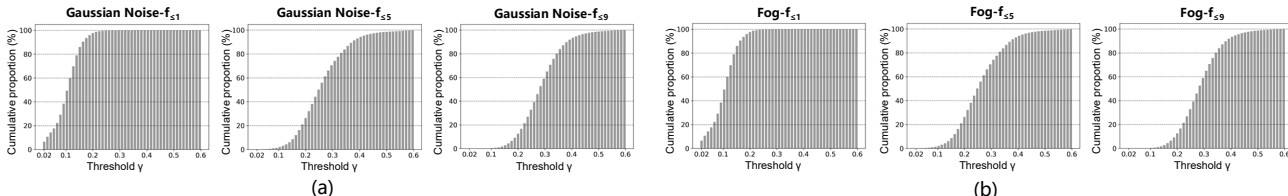

*Figure 2.* The cumulative distribution of $\gamma$-robust features at different sub-networks under the corruption of (a) Gaussian Noise and (b) Fog. The consistent shift toward higher thresholds in deeper layers indicates a progressive scarcity of robust features as the network depth increases, revealing that high-level representations are more sensitive to structured corruptions.

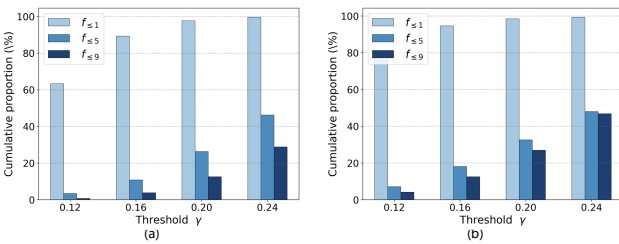

*Figure 3.* Robust feature proportions across thresholds $\gamma$ under (a) Gaussian Noise and (b) Fog. The downward trend across stages ($f_{\leq 1} \rightarrow f_{\leq 9}$) reveals systemic robustness decay.

# 3. Methodology

## 3.1. Problem Formulation

We consider the robustness of a neural network $f_\theta$ under natural image corruptions. Formally, let $\mathcal{D} = \{(x_i, y_i)\}_{i=1}^{N}$ be a clean dataset from $\mathbb{P}(X, Y)$. A corruption function $h \in \mathcal{H}$ transforms $x_i$ into a corrupted version $h(x_i)$. Following the covariate shift assumption (Gavrikov & Keuper, 2024), where $\mathbb{P}(X) \neq \mathbb{P}(h(X))$ but the label remains invariant, our objective is to minimize the expected risk across the corruption space $\mathcal{H}$:

$$\min_\theta \mathbb{E}_{h \sim H}[\mathbb{E}_{X,Y \sim \mathbb{P}(X,Y)}[\mathcal{L}(f_\theta(h(X)), Y)]], \quad (1)$$

where $\mathcal{L}$ denotes a standard supervised loss (e.g., cross-entropy). Unlike adversarial robustness works that emphasize input-level sensitivity (Ilyas et al., 2019), we posit that for natural corruptions, robustness is fundamentally tied to the stability of internal features. To formalize this, we decompose $f_\theta$ into $L$ layers: $f_\theta(\cdot) = f_L \circ f_{L-1} \circ \cdots \circ f_1$, where $f_{\leq l}$ maps the input into a feature map $Z_l$. Each neuron in layer $l$ constitutes a **computational pathway** $P_{l,k} : X \rightarrow Z_{l,k}$ (abbreviated as $P$). The robustness of $f_\theta$ thus depends on whether these pathways can produce invariant features $Z_{l,k}$ under corruption $h$.

## 3.2. Robust and Non-robust Features

### 3.2.1. CHARACTERIZING ROBUST FEATURES AND PATHWAYS

The concept of robust features was introduced by (Ilyas et al., 2019) via a pixel-level disentanglement framework to interpret adversarial examples. Subsequent studies extended this analysis to the layer (Yan et al., 2021) and neuron levels (Zhang et al., 2020; Madaan et al., 2020) to enhance adversarial robustness. However, while adversarial perturbations are typically high-dimensional, non-semantic noise affecting the local pixel manifold, natural corruptions (e.g., fog, blur) exhibit global structural patterns that manifest as semantic shifts within the latent space. Despite this, the nature of robust and non-robust features under such structured distortions remains poorly understood, particularly regarding their explicit disentanglement. To fill this gap, we formalize robust features under image corruptions through the lens of computational pathways.

**Definition 3.1** (Robust features). Given a normalized distortion metric $D$, a feature $z_{l,k}$ is said to be $\gamma$-robust ($\gamma > 0$) for corruption $h$ if

$$D(z_{l,k}, \hat{z}_{l,k}) < \gamma, \quad (2)$$

where $\hat{z}_{l,k}$ denotes the corrupted version of $z_{l,k}$ induced by applying the corruption $h$ to the input.

**Definition 3.2** (Robust pathways). Given a normalized distortion metric $D$, a computational pathway $P$ is said to be $\gamma$-robust ($\gamma > 0$) for corruption $h$ if

$$\mathbb{E}_{x \sim \mathbb{P}(X)}[D(P(x), P(h(x)))] < \gamma. \quad (3)$$

Note that these definitions isolate robustness as a property of features and pathways, independent of their impact on predictions; Section 3.2.3 later investigates this connection.

### 3.2.2. IDENTIFYING ROBUST FEATURES: LAYER-WISE DISTRIBUTION AND EVOLUTION

To demystify the internal behavior of robust features, we quantify their layer-wise distribution across the network. We employ the local Lipschitz constant (Cisse et al., 2017;

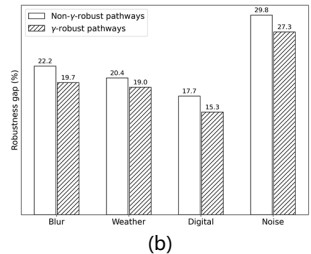 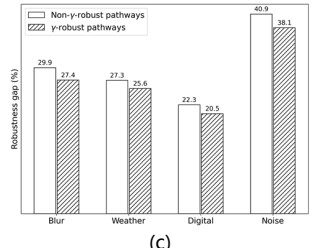 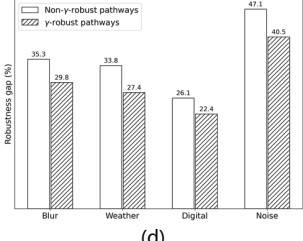

*Figure 4.* Impact of pathway robustness on sub-networks and the full model. We report the robustness gap (lower is better) for (a) $f_{\leq 1}$, (b) $f_{\leq 5}$, (c) $f_{\leq 9}$, and (d) $f_\theta$ when preserving either $\gamma$-robust or non-$\gamma$-robust pathways.

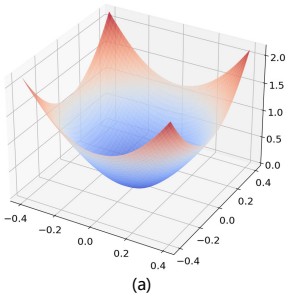 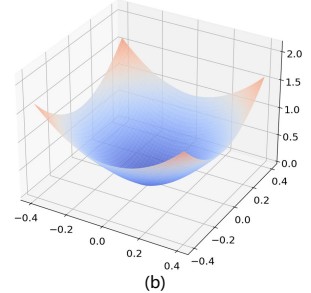

*Figure 5.* Loss landscapes of $f_\theta$ under input perturbations. (a) Non-$\gamma$-robust pathways exhibit a sharp landscape, whereas (b) $\gamma$-robust pathways show a flatter surface.

Szegedy et al., 2013) as the distortion metric $D$ in Equation (2), denoted as $D_{\text{Lip}}$:

$$D_{\text{Lip}}(z_{l,k}, \hat{z}_{l,k}) = \frac{||z_{l,k} - \hat{z}_{l,k}||}{||x - h(x)||}, \qquad (4)$$

where $|| \cdot ||$ represents the normalized Frobenius norm. This metric effectively captures the sensitivity of internal features relative to input-level perturbations. Using Equation (2), Figure 2 (a)-(b) visualize the cumulative distribution of $\gamma$-robust features for a pretrained ResNet18 model under Gaussian noise and Fog, respectively. To provide a more intuitive and quantitative comparison, Figure 3 summarizes the proportion of robust features at specific thresholds $\gamma$.

Our empirical analysis reveals a non-homogeneous evolution of robustness along the network depth. Specifically, while shallow sub-networks (e.g., $f_{\leq 1}$) retain a high proportion of $\gamma$-robust features, this proportion exhibits a progressive decay as information propagates into deeper sub-networks (e.g., $f_{\leq 5}$ and $f_{\leq 9}$). As evidenced by the sharp contrast in Figure 3, the available pool of robust features shrinks drastically in deeper layers, particularly at stringent thresholds. This trend suggests that as receptive fields expand to capture global structural patterns, features become increasingly susceptible to semantic shifts induced by structured corruptions. Consistent results for additional corruption types are further detailed in Appendix D.1, underscoring that internal robustness degradation is a pervasive

challenge across diverse distortion categories.

*Remark* 1. Under image corruptions, robust features exhibit a progressive decay across neural layers, indicating that their prevalence diminishes significantly as features propagate into deeper stages of the network.

### 3.2.3. A CLOSER LOOK AT ROBUST FEATURES: CAUSAL IMPACT AND LOSS LANDSCAPES

Based on the above observation, we take a closer look at the collective behavior of robust (non-robust) features under image corruptions. To explore this, we investigate the causal role of these features by isolating robust and non-robust pathways and quantifying their impact on the robustness of both downstream sub-networks and the full model. For a given corruption $h$, we score each computational pathway $P$ in the stem stage ($l_1$) of a pretrained ResNet-18 using the empirical mean of $D_{\text{Lip}}$:

$$\bar{D}_{\text{Lip}}(P, h) = \frac{1}{N} \sum_{i=1}^{N} \frac{||P(x_i) - P(h(x_i))||}{||x_i - h(x_i)||}. \qquad (5)$$

To evaluate their causal role, we partition pathways into $\gamma$-robust and non-$\gamma$-robust groups based on the top and bottom 50% of scores. We selectively activate one group at layer $l_1$ while zeroing the other. For a fair comparison, we perform linear probing by attaching and fine-tuning a linear head on the final feature maps of each sub-network $f_{\leq l_2}$ and the full model $f_\theta$ using clean data. Comprehensive analysis across 15 corruption types in Appendix D.2 further confirms the consistent advantage of $\gamma$-robust pathways.

We evaluate the influence of these pathway groups using the robustness gap (Taori et al., 2020; Fang et al., 2022; Tu et al., 2025), defined as the accuracy degradation from clean to corrupted data. Figure 4 reveals a consistent trend across all sub-networks and the full model: $\gamma$-robust pathways yield a significantly narrower gap compared to their non-robust counterparts. This disparity suggests that model robustness is sustained by this specific subset of invariant features. To provide a geometric explanation, we visualize the loss landscapes of $f_\theta$ in Figure 5. Notably, $\gamma$-robust pathways exhibit a significantly flatter loss surface, which

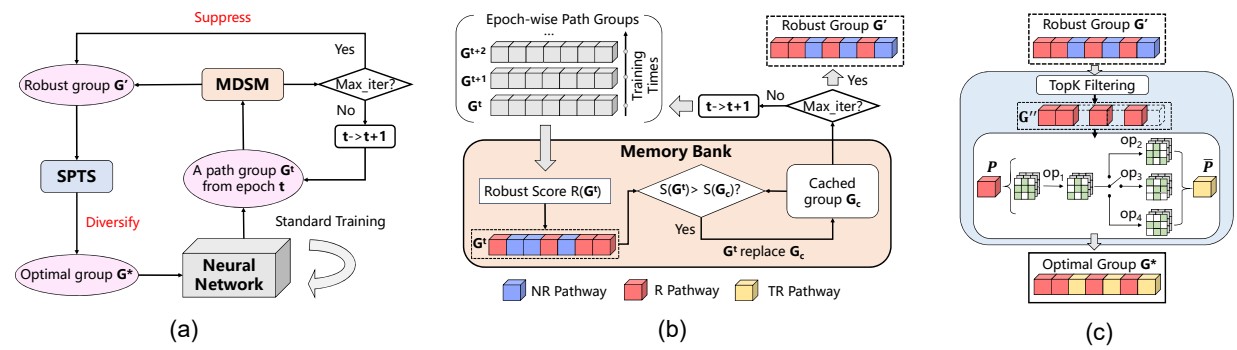

*Figure 6.* The pipeline of (a) S&D, (b) MDSM, and (c) SPTS. NR, R and TR denote non-robust, robust and tweaked robust pathways, respectively.

ensures stable predictions in the vicinity of clean inputs and effectively minimizes the robustness gap.

*Remark* 2. The robustness of the full model is functionally dependent on the prevalence of robust features, which exert a direct impact on preserving performance across downstream layers and sub-networks under corruptions.

### 3.3. The S&D Refinement

#### 3.3.1. OVERVIEW

Motivated by the hierarchical decay of robust features, we propose the S&D refinement to enhance corruption robustness through pathway manipulation (Figure 6 (a)). S&D consists of two synergistic components: (1) MDSM, which identifies a stable pathway group $G'$ generating robust representations, and (2) SPTS, which constructs an optimal group $G^*$ by diversifying $G'$ while preserving its structural invariants. Notably, S&D is non-intrusive, requiring no architectural modifications or interference with standard training, ensuring seamless integration into mainstream frameworks.

#### 3.3.2. MDSM: SELECTING ROBUST PATHWAYS

The Memory-aware Dynamic Selection Mechanism (MDSM) identifies robust pathways by maintaining a candidate group $G_c$ in a memory bank (Figure 6 (b)). At epoch $t$, a candidate collection $G^t = \{P_{l,1}, \ldots, P_{l,K}\}$ is deterministically constructed from the current model state, with fixed indexing inherited from layer $l$. This collection is evaluated against $G_c$ using a fitness score $\mathcal{S}$ that balances feature stability with discriminability:

$$\mathcal{S}(G^t) = \frac{1}{K} \sum_{k=1}^{K} [R(P_{l,k}(x), P_{l,k}(\hat{x}))] + \lambda \cdot \sigma(n/N), \quad (6)$$

where $X$ and $\hat{X}$ represent a batch of clean images and their corresponding corrupted versions. $R$ is a similarity metric (e.g., $L_2$ or CKA), and $n$ and $N$ denote the current and total scheduled MDSM steps at the epoch level. $\lambda$ is a hyperparameter balancing two terms and is set to 1. The second

term, governed by a sigmoid-like schedule $\sigma(\cdot)$, prioritizes discriminability as training matures, inspired by findings that deeper layers develop more discriminative representations over time (Zeiler & Fergus, 2014; Yosinski et al., 2014; Zhang et al., 2021). Using Equation (6), the memory bank iteratively updates $G_c$ over $N$ steps to output the robust pathway group $G'$. Ablations in Appendix B.4 and B.3 confirm that S&D yields consistent gains regardless of the specific choice of $R$ or the transformations for $\hat{x}$. Notably, MDSM operates during epoch gaps, ensuring zero interference with standard training gradients.

#### 3.3.3. SPTS: DIVERSIFYING ROBUST PATHWAYS

To prevent overfitting to specific robust patterns, we propose the Structure-consistent Pathway Tweaking Strategy (SPTS) to diversify $G'$. As illustrated in Figure 6 (c), SPTS first filters the top-$K$ most robust pathways:

$$G'' = \text{TopK}(G', R), \quad (7)$$

where $\text{TopK}(\cdot)$ denotes the operation that retains the $K$ pathways in $G'$ according to their $R$ values, yielding the new group $G''$. $K$ is typically set to $|G'|/2$; the impact of varying $K$ is detailed in Appendix B.5. To restore the group capacity, we generate augmented pathways $\bar{P}$ by applying symmetry-preserving transformations to the final weight matrix $A \in \mathbb{R}^{C \times W \times H}$ of pathways in $G''$:

$$\bar{A} = \text{op}_i(\text{op}_1(A)) \quad (8)$$
$$\text{subject to} \quad \text{op}_i \sim U\{\text{op}_2, \text{op}_3, \text{op}_4\},$$

where $U$ denotes uniform sampling from a predefined set of operations. The transformation suite includes: `Channel shuffling` ($\text{op}_1$), `Horizontal flipping` ($\text{op}_2$), `Vertical flipping` ($\text{op}_3$), and `Matrix transpose` ($\text{op}_4$), with implementation details in Appendix A.1. This specific combination was identified as a highly effective configuration through comparative experiments in Appendix B.2. The augmented pathways $\bar{P}$ (parameterized by $\bar{A}$) introduce stochastic variations into

*Table 1.* We report Top-1 Accuracy (↑) for ImageNet and mean Corruption Error (mCE) for its corrupted variants (C, C̄, 3DCC, and V2-C). Avg.mCE (↓) averages these four error rates. [·] indicates ensemble-based results; (+) and (−) denote performance changes via S&D. * indicates the use of modern training recipes, including stronger augmentation and optimization practices.

| Main | S&D | ImageNet | ImageNet-C | ImageNet-C̄ | ImageNet-3DCC | ImageNetV2-C | Avg.mCE (↓) |
|---|---|---|---|---|---|---|---|
| | | | Backbones of Neural Network | | | | |
| MobileNetV2 (Sandler et al., 2018) | ✗ | 71.4 | 86.2 | 86.4 | 82.8 | 89.8 | 86.3 |
| | ✔ | 71.6 [71.2] | 84.9 | 86.6 | 82.6 | 89.2 | 85.8 (**-0.5**) |
| ResNet-50 (He et al., 2016) | ✗ | 75.7 | 76.7 | 79.4 | 73.2 | 83.1 | 78.1 |
| | ✔ | 74.5 [75.6] | 72.4 | 78.2 | 71.3 | 79.2 | 75.3 (**-2.8**) |
| VGG19 (Simonyan, 2014) | ✗ | 73.9 | 81.6 | 84.6 | 78.6 | 86.4 | 82.8 |
| | ✔ | 75.9 [74.2] | 79.7 | 81.1 | 76.9 | 84.9 | 80.7 (**-2.1**) |
| WideResNet-50_2 (Zagoruyko, 2016) | ✗ | 78.3 | 71.7 | 74.4 | 68.9 | 79.2 | 73.6 |
| | ✔ | 76.3 [78.3] | 68.5 | 73.9 | 67.7 | 76.1 | 71.6 (**-2.0**) |
| | | | Data Augmentation & Model Regularization | | | | |
| AugMix (Hendrycks et al., 2019) | ✗ | 76.2 | 72.8 | 74.8 | 70.3 | 79.6 | 74.4 |
| | ✔ | 74.9 [76.2] | 68.1 | 75.4 | 68.3 | 75.7 | 71.9 (**-2.5**) |
| TriAug (Müller & Hutter, 2021) | ✗ | 76.7 | 70.3 | 75.5 | 67.5 | 77.6 | 72.7 |
| | ✔ | 75.3 [76.7] | 67.7 | 75.7 | 66.3 | 75.4 | 71.3 (**-1.4**) |
| Label Smoothing (Müller et al., 2019) | ✗ | 76.6 | 75.2 | 77.1 | 72.1 | 81.5 | 76.5 |
| | ✔ | 75.0 [76.6] | 72.1 | 77.3 | 70.8 | 79.0 | 74.8 (**-1.7**) |
| Mixup (Zhang et al., 2018) | ✗ | 76.7 | 71.7 | 71.4 | 70.1 | 78.7 | 73.0 |
| | ✔ | 74.9 [76.7] | 68.1 | 71.1 | 68.8 | 75.6 | 70.9 (**-2.1**) |
| | | | Advanced Training Recipe | | | | |
| MobileNetV2* (Sandler et al., 2018) | ✗ | 72.1 | 86.1 | 82.0 | 81.3 | 89.2 | 84.7 |
| | ✔ | 72.5 [72.3] | 78.5 | 78.3 | 76.3 | 83.7 | 79.2 (**-5.5**) |
| ResNet-50* (He et al., 2016) | ✗ | 80.7 | 66.5 | 66.1 | 62.3 | 73.7 | 67.2 |
| | ✔ | 79.5 [79.7] | 61.5 | 65.0 | 60.9 | 69.8 | 64.3 (**-2.9**) |
| ConvNeXt-base* (Liu et al., 2022) | ✗ | 84.0 | 53.6 | 53.1 | 53.1 | 63.5 | 55.8 |
| | ✔ | 82.9 [83.4] | 50.7 | 50.1 | 51.8 | 61.2 | 53.5 (**-2.3**) |

*Table 2.* Comparison with robust learning paradigms on ImageNet-C. We report Top-1 Accuracy (↑). * denotes methods that used either strong augmentation (S&D*) or improved training recipes (AdaSAP$_P^*$).

| Baseline | Stochastic Depth | AdaSAP$_P^*$ | EWS |
|---|---|---|---|
| 39.2 | 38.9 (-0.3) | 43.3 (+4.1) | 40.6 (+1.4) |
| **DST** | **DAMP** | **DAT** | **TVM** |
| 38.7 (-0.5) | 41.4 (+2.2) | 41.1 (+1.9) | 39.8 (+0.6) |
| **VOneNet** | **GaborNet** | **S&D** | **S&D*** |
| 40.3 (+1.1) | 37.5 (-1.7) | 42.7 (+3.5) | 46.3 (+7.1) |

the latent space while adhering to the robust hierarchical constraints of the original group. By merging $G''$ with these augmented pathways, we form the optimal group $G^*$. Finally, $G^*$ is integrated into the model $f_\theta$ for standard training, ensuring that the network is guided by a diverse yet stable set of computational pathways.

## 4. Experiments

### 4.1. Experimental Setup

**Evaluation dataset.** The evaluation uses eight robustness benchmark datasets. For **image classification**, we use ImageNet-C (Hendrycks & Dietterich, 2019), ImageNet-C̄ (Mintun et al., 2021), and ImageNet-3DCC (Kar et al., 2022). ImageNet-C includes 15 corruption types at five severity levels, ImageNet-C̄ adds 10 new corruptions, and ImageNet-3DCC introduces 12 3D-based corruptions mimicking real-world distortions. Additionally, we create ImageNetV2-C by applying 15 corruption types to ImageNetV2 (Recht et al., 2019). For **object detection**, we use COCO-C (Michaelis et al., 2019), a corrupted version of COCO (Lin et al., 2014). For **semantic segmenta-**

**tion**, we generate VOC-C, Cityscapes-C, and ADE20K-C from PASCAL VOC 2012 (Everingham et al., 2015), Cityscapes (Cordts et al., 2016), and ADE20K (Zhou et al., 2019). All datasets are used exclusively for testing.

**Evaluation design.** We compare S&D with: (1) Sub-network selection (Stochastic Depth (Huang et al., 2016), AdaSAP (Bair et al., 2024)); (2) Dynamic weight evolution (EWS (Guo et al., 2022), DST (Wu et al., 2025), DAMP (Trinh et al., 2024)); (3) Consistency learning (DAT (Mao et al., 2022), TVM (Saikia et al., 2021)); and (4) Bio-inspired design (VOneNet (Dapello et al., 2020), Gabor-Net (Pérez et al., 2020)). To demonstrate broad applicability, we evaluate S&D across classification, detection, and segmentation using CNN and Transformer backbones, combined with data augmentations, regularization, and widely adopted training recipes (detailed in Appendix Table A.1). We further assess robustness under test-time adaptation and real-world corruptions. For fairness, all models follow standard protocols or their original settings. Metrics include clean accuracy/mAP/mIoU and mCE or average corrupted performance. Full baseline details are provided in Appendix A.3.

**Implementation details.** S&D is applied during the initial training phase ($N = 5$ in Equation (6)). We update the stem stage of each backbone as the updated pathway group $G^t$, a choice validated against deeper stages in Appendix B.6. We adopt CKA (Kornblith et al., 2019) as the default similarity metric $R$, which yields consistent gains across various metric alternatives (Appendix B.4). We also explore an ensemble-based strategy for evaluating the clean images (Appendix A.2). All classification models follow

*Table 3.* We report Top-1 Accuracy (↑) for ImageNet-100 and mean Corruption Error (mCE) for its variants (C, $\bar{C}$, 3DCC, and V2-C). Avg. mCE (↓) denotes the average error across these four benchmarks. (+) and (−) reflect the performance changes after applying S&D.

| Main | S&D | IN-100 | IN-100-C | IN-100-$\bar{C}$ | IN-100-3DCC | IN-100V2-C | Avg.mCE (↓) |
|---|---|---|---|---|---|---|---|
| MobileViT_S | ✗ | 85.4 | 88.8 | 92.5 | 84.8 | 92.9 | 89.8 |
| | ✔ | 85.7 | 87.3 | 91.6 | 85.1 | 91.6 | 88.9 (**-0.9**) |
| EfficientFormer_L1 | ✗ | 91.6 | 73.7 | 66.3 | 65.2 | 80.4 | 71.4 |
| | ✔ | 91.8 | 74.4 | 63.4 | 64.0 | 80.6 | 70.6 (**-0.8**) |
| ViT_Tiny | ✗ | 84.5 | 76.2 | 72.7 | 78.2 | 81.7 | 77.2 |
| | ✔ | 86.3 | 76.1 | 70.5 | 77.8 | 80.3 | 76.2 (**-1.0**) |
| Mambaout_femto | ✗ | 94.1 | 60.0 | 54.7 | 57.7 | 68.2 | 60.2 |
| | ✔ | 94.2 | 57.6 | 56.3 | 57.5 | 67.4 | 59.7 (**-0.5**) |

*Table 4.* The semantic segmentation performance on ADE20K-C, Cityscapes-C and VOC-C. We report the mean Intersection over Union (mIoU ↑). S and R represent the results on clean and corrupted data, respectively.

| Main | S&D | ADE20K-C | | Cityscapes-C | | VOC-C | |
|---|---|---|---|---|---|---|---|
| | | S | R | S | R | S | R |
| | | | | Multi-scale Fusion-based Structure | | | |
| DeepLabV3+ (Chen et al., 2018) | ✗ | 42.1 | 21.3 | 78.6 | 36.2 | 75.9 | 44.9 |
| | ✔ | 41.2 | 23.1 (**+1.8**) | 77.9 | 39.2 (**+3.0**) | 75.4 | 48.0 (**+3.1**) |
| PSPNet (Zhao et al., 2017) | ✗ | 41.2 | 20.1 | 77.2 | 34.3 | 75.9 | 44.5 |
| | ✔ | 40.0 | 21.5 (**+1.4**) | 76.3 | 37.8 (**+3.5**) | 76.0 | 47.4 (**+2.9**) |
| | | | | Attention-based Structure | | | |
| Mask2Former (Cheng et al., 2022) | ✗ | 47.2 | 25.2 | 79.3 | 40.8 | – | – |
| | ✔ | 46.4 | 29.2 (**+4.0**) | 77.3 | 47.1 (**+6.3**) | – | – |
| ANN (Zhu et al., 2019) | ✗ | 39.6 | 19.2 | 76.5 | 33.6 | 74.9 | 43.3 |
| | ✔ | 38.9 | 20.6 (**+1.4**) | 75.5 | 35.9 (**+2.3**) | 75.0 | 46.7 (**+3.4**) |
| CCNet (Huang et al., 2019) | ✗ | 41.2 | 20.1 | 78.9 | 34.0 | 76.2 | 44.1 |
| | ✔ | 40.2 | 21.5 (**+1.4**) | 78.0 | 38.0 (**+4.0**) | 75.9 | 47.6 (**+3.5**) |
| DANet (Fu et al., 2019) | ✗ | 40.1 | 19.9 | 78.0 | 34.5 | 74.5 | 43.9 |
| | ✔ | 39.4 | 22.0 (**+2.1**) | 77.7 | 38.5 (**+4.0**) | 74.2 | 46.6 (**+2.7**) |
| GCNet (Cao et al., 2019) | ✗ | 40.3 | 19.9 | 76.8 | 33.1 | 76.4 | 43.9 |
| | ✔ | 39.5 | 21.6 (**+1.7**) | 76.5 | 36.6 (**+3.5**) | 74.8 | 47.5 (**+3.6**) |
| PSANet (Zhao et al., 2018) | ✗ | 40.8 | 20.0 | 77.0 | 34.3 | 76.4 | 44.6 |
| | ✔ | 40.1 | 21.6 (**+1.6**) | 75.9 | 36.7 (**+2.4**) | 75.5 | 46.6 (**+2.0**) |

*Table 5.* The object detection performance on COCO and COCO-C. We report the mean Intersection over Union (mAP ↑).

| Main | S&D | COCO | COCO-C |
|---|---|---|---|
| | | Two-Stage Detector | |
| Faster RCNN (Ren et al., 2016) | ✗ | 37.6 | 17.5 |
| | ✔ | 37.2 | 18.2 (**+0.7**) |
| Mask RCNN (He et al., 2017) | ✗ | 38.1 | 18.0 |
| | ✔ | 38.1 | 18.5 (**+0.5**) |
| Cascade RCNN (Cai & Vasconcelos, 2018) | ✗ | 40.5 | 18.9 |
| | ✔ | 40.5 | 19.5 (**+0.6**) |
| Cascade Mask RCNN (Cai & Vasconcelos, 2018) | ✗ | 41.1 | 19.3 |
| | ✔ | 41.1 | 19.9 (**+0.6**) |
| | | One-Stage Detector | |
| RetinaNet (Ross & Dollár, 2017) | ✗ | 36.3 | 17.1 |
| | ✔ | 36.6 | 17.6 (**+0.5**) |
| YOLOv5s (Reis et al., 2023) | ✗ | 39.8 | 19.9 |
| | ✔ | 39.1 | 22.1 (**+2.2**) |
| YOLOv8s (Reis et al., 2023) | ✗ | 46.1 | 25.5 |
| | ✔ | 45.0 | 27.0 (**+1.5**) |

standard PyTorch recipes, while detection and segmentation tasks utilize ImageNet-pretrained ResNet-50 via MMDetection (Chen et al., 2019) and MMSegmentation (Contributors, 2020). Ablation studies regarding transformation types and hyper-parameter sensitivity are detailed in Appendix B.

## 4.2. Experimental Results

### 4.2.1. MAIN RESULTS ON ROBUST CLASSIFICATION

**Comparison with baselines.** We evaluate S&D against diverse robust training paradigms: sub-network selection, weight evolution, consistency learning, and bio-inspired designs (Table 2). S&D consistently achieves superior robust

accuracy. Without strong augmentation, S&D surpasses the second-best method, DAMP, by 1.3%. When combined with strong augmentation (*), S&D yields a 7.1% total gain over the baseline and outperforms AdaSAP$_P^*$ by 3.0%. This confirms S&D's effectiveness as a standalone strategy. For a fair comparison, results are compiled from original reports or our implementations; see Appendix A.3 for details.

**Compatibility with training recipes.** Table 1 shows S&D's efficacy across various backbones and configurations. We observe consistent gains of 0.4–2.5% regardless of model scale. Notably, S&D exhibits strong synergy with augmentations and regularizations, boosting robust accuracy by 2.2% (AugMix) and 1.9% (Mixup). Furthermore, under modern recipes, S&D yields substantial gains of 4.3% on MobileNetV2 and 2.0% on ConvNeXt-Base. These results confirm S&D as a versatile plug-and-play enhancement; see Appendix C.1 for extended results.

### 4.2.2. VERSATILITY ACROSS ARCHITECTURES, TASKS AND SCENARIOS

**Architecture diversity.** We evaluate S&D on Transformer and Mamba backbones using ImageNet-100 and its corruption variants (e.g., -C, -$\bar{C}$, -3DCC). S&D consistently enhances robustness across these diverse architectures (Table 3). While gains are most pronounced for CNNs due to their texture bias (Geirhos et al., 2018a), S&D still yields

*Table 6.* The segmentation performance (mIOU) on ACDC.

| Main | S&D | Fog | Night | Rain | Snow | Avg. |
|---|---|---|---|---|---|---|
| DeepLabV3+ | ✗ | 63.3 | 13.3 | 49.2 | 46.0 | 43.0 |
| | ✔ | 64.1 | 11.0 | 49.5 | 47.8 | 43.1 (**+0.1**) |
| PSPNet | ✗ | 62.0 | 9.7 | 45.2 | 39.8 | 39.2 |
| | ✔ | 62.8 | 10.9 | 46.7 | 42.7 | 40.8 (**+1.6**) |
| Mask2Former | ✗ | 63.0 | 25.5 | 49.8 | 50.0 | 47.1 |
| | ✔ | 64.0 | 23.7 | 48.3 | 51.4 | 46.9 (**-0.2**) |
| CCNet | ✗ | 61.4 | 8.7 | 49.9 | 45.8 | 41.5 |
| | ✔ | 63.6 | 13.9 | 44.5 | 42.9 | 41.2 (**-0.3**) |
| DANet | ✗ | 61.1 | 9.2 | 46.3 | 43.6 | 40.1 |
| | ✔ | 61.1 | 10.0 | 48.6 | 41.2 | 40.2 (**+0.1**) |
| GCNet | ✗ | 57.7 | 5.6 | 45.1 | 36.8 | 36.3 |
| | ✔ | 61.5 | 10.7 | 45.6 | 42.7 | 40.1 (**+3.8**) |

*Table 7.* Comparison of test-time adaptation Methods at the severity 5 on ImageNet-C (Without vs. With S&D). We report Top-1 Accuracy ($\uparrow$).

| | AdaContrast | Tent | BN | EATA |
|---|---|---|---|---|
| w/o S&D | 34.9 | 37.3 | 31.5 | 42.1 |
| w/ S&D | 38.1 (+3.2) | 38.3 (+1.0) | 31.9 (+0.4) | 42.6 (+0.5) |
| | SANTA | SAR | RMT | ROTTA |
| w/o S&D | 39.9 | 37.8 | 42.2 | 32.6 |
| w/ S&D | 40.6 (+0.7) | 38.6 (+0.8) | 44.2 (+2.0) | 34.0 (+1.4) |

steady improvements for ViT and Mamba models. These results demonstrate S&D's versatility across various architectural paradigms.

**Downstream tasks.** To verify S&D's versatility, we extend evaluation to dense prediction tasks. For semantic segmentation (Table 4), S&D yields consistent mIoU gains across ADE20K-C (1.4–4.0%), Cityscapes-C (2.3–6.3%), and VOC-C (2.0–3.6%), with Mask2Former achieving the largest boost. In object detection (Table 5), S&D effectively enhances robust mAP for both two-stage (Faster R-CNN: 17.5→18.2) and one-stage detectors (YOLOv5s: 19.9→22.1) while maintaining competitive clean performance. These results confirm S&D's broad applicability in enhancing robustness for diverse visual applications.

**Real-world scenarios.** We first validate S&D under real-world adverse weather for semantic segmentation and object detection. On the ACDC dataset (Sakaridis et al., 2021), S&D improves mIoU across most backbones (Table 6), notably boosting GCNet by 3.8%, with corresponding segmentation visualizations provided in Appendix D.3. On the DWD benchmark (Wu & Deng, 2022), S&D enhances Faster R-CNN by 1.6 and 0.4 mAP in night-rainy and dusk-rainy scenarios, respectively. Qualitative results (Figure 7) confirm S&D's efficacy in localizing and identifying objects under such compound corruptions. To further assess S&D in dynamic environments, we evaluate continual test-time adaptation (TTA) on ImageNet-C. By integrating S&D with eight representative methods, including Tent (Wang et al., 2021), BN (Benz et al., 2021), AdaContrast (Chen et al., 2022), EATA (Niu et al., 2022), SANTA (Chakrabarty et al., 2023), SAR (Niu et al., 2023), RMT (Döbler et al., 2023), and ROTTA (Yuan et al., 2023), we observe consistent performance gains (Table 7) with a 3.2% peak improvement.

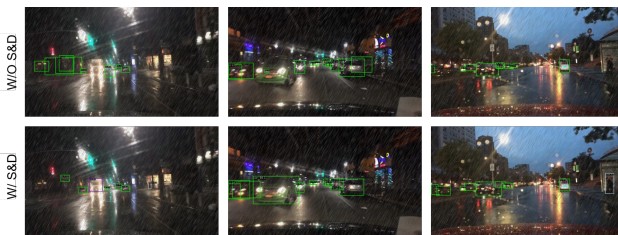

*Figure 7.* The detection results in night-rainy scenarios.

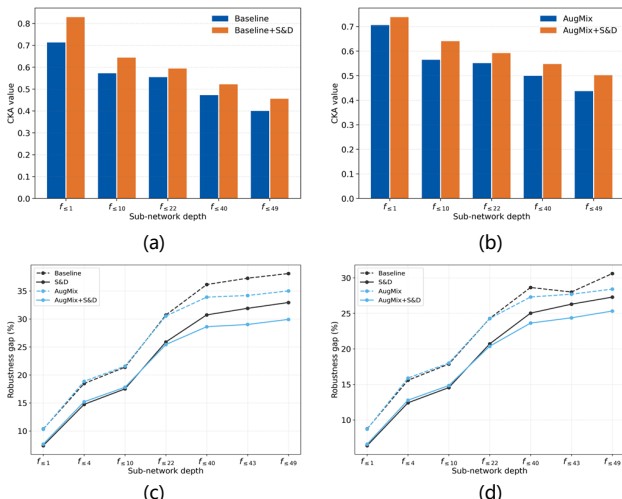

*Figure 8.* Comparison of (a) CKA between the baseline and S&D, (b) CKA between AugMix and AugMix+S&D, and robustness gap with and without S&D on (c) ImageNet-C and (d) ImageNet-3DCC.

These results demonstrate S&D's broad compatibility in complex, real-world degraded environments.

## 5. Analysis

### 5.1. Representation Stability

To evaluate representation stability, we compute CKA between ResNet-50 sub-network features ($f_{\leq 1}$ to $f_{\leq 49}$) under clean and corrupted inputs. As shown in Figure 8 (a, b), S&D consistently increases feature similarity across all depths. Further linear probing on ImageNet-C/3DCC using the robustness gap (lower is better) reveals consistent gap reductions in Figure 8 (c, d). These findings confirm that S&D effectively guides the network toward robust hierarchical representations in a bottom-up manner.

### 5.2. Training Efficiency

We measured the training-time and memory overhead of S&D relative to a standard ResNet-50 baseline. Table 8 shows that baseline training requires 2294.27 s/epoch, while MDSM adds only 0.3233 s on selected epochs ($\sim$0.01%), and the one-time SPTS cost is 0.0057 s ($<$0.001%), with

*Table 8.* Wall-clock overhead analysis of our method. "Baseline" denotes the runtime of the original training per epoch. "MDSM" denotes the additional per-epoch overhead introduced by our method, and "SPTS" is a one-time operation. Within MDSM, "Aug" denotes corrupted input generation, "Fwd" denotes forward propagation, and "CKA" denotes CKA computation. Relative overhead is measured with respect to the baseline epoch time.

| Component | Execution Frequency | Time (s) | Time Breakdown (s) | Against Baseline Epoch |
|---|---|---|---|---|
| Baseline training | Every epoch | 2294.27 | – | – |
| MDSM | Selected epochs | $0.3233 \pm 0.0015$ | Aug: 0.0056, Fwd: 0.0420, CKA: 0.2757 | ~0.01% |
| SPTS | Last selected epoch | $0.0057 \pm 0.0070$ | – | $< 0.001\%$ |

*Table 9.* Throughput and memory overhead analysis of MDSM compared with baseline training. "Peak Mem Diff" is computed as MDSM peak memory minus baseline peak memory. Negative values indicate that MDSM remains within the baseline training memory budget.

| Baseline Throughput (img/s) | MDSM Throughput (img/s) | Baseline Peak Mem (MB) | MDSM Peak Mem (MB) | Peak Mem Diff (MB) |
|---|---|---|---|---|
| 880.79 | 6003.23 | 21292.07 | 7640.79 | -13651.28 |

CKA computation dominating the small overhead. Table 9 reports peak GPU memory and throughput: MDSM consumes 7640.79 MB versus 21292.07 MB for baseline, while throughput increases due to lighter memory usage. Overall, S&D introduces negligible wall-clock and memory overhead during training, confirming its practical efficiency without affecting test-time cost.

## 6. Conclusion

This paper investigates the phenomenon of robust features under image corruptions. Our analysis reveals that robust features exhibit a progressive decay across neural layers and establishes that model-wide robustness is functionally dependent on their prevalence. Building on these insights, we propose S&D, a simple refinement that suppresses non-robust pathways and diversifies robust ones during training. S&D is architecture-agnostic, parameter-free, and requires zero test-time overhead, enabling seamless plug-and-play integration. Extensive experiments demonstrate that S&D significantly enhances corruption robustness and OOD generalization, offering a new perspective on building robust vision models.

Despite these improvements, several limitations remain. S&D introduces a consistent clean–corruption trade-off, as pathways stabilized under corruptions may suppress features beneficial for clean accuracy. Evaluations have focused on moderate-scale models and datasets; scalability to larger models and full ImageNet-scale data remains to be fully explored. The current framework targets 2D/3D vision tasks, and applicability to other modalities is not yet established. Future work will focus on developing a theoretical framework for internal robust features, designing strategies to mitigate clean-accuracy degradation, and extending S&D to additional modalities such as audio and multimodal tasks.

## Impact Statement

We propose Suppress & Diversify (S&D), a training-time refinement that enhances the robustness of deep neural networks under natural image corruptions. S&D improves reliability in safety-critical vision systems such as autonomous driving, surveillance, and medical imaging, without adding inference overhead. It is architecture-agnostic and compatible with diverse CNN and Transformer backbones. Potential risks include over-reliance on automated decisions in untested scenarios; users should validate models across varied environments and combine with augmentation or uncertainty estimation. S&D provides a principled approach to building reliable and generalizable visual models.

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

# A. Implementation Details

## A.1. The Tweaking Operations

The tweaking operations include four types, Channel shuffling, Horizontal flipping, Vertical flipping, Matrix transpose, abbreviated as op1, op2, op3 and op4. Let $A \in \mathbb{R}^{C \times W \times H}$ be the weight matrix, and $c, w, h$ are the corresponding index, respectively. The tweaking operations can be formulated as:

$$op_i(x) = \begin{cases} A_{\sigma(c),w,h} & \text{if } i = 1 \\ A_{c,W-w+1,h} & \text{if } i = 2 \\ A_{c,w,H-h+1} & \text{if } i = 3 \\ A_{c,h,w} & \text{if } i = 4, \end{cases} \quad \text{(A.1)}$$

where $\sigma$ is a random permutation that rearranges the C channels.

## A.2. Ensemble-based Strategy

We additionally test a simple ensemble-based strategy on ImageNet. Specifically, for Model A, we select the ImageNet-pretrained model provided by PyTorch, while Model B uses the same architecture but incorporates S&D, such as ResNet-50 and ResNet-50+S&D. Inspired by (Park et al., 2023), we find that the $l_2$ norm of the output at the first layer can serve as a confidence value to distinguish between non-corrupted and corrupted data. Given an unknown sample $x$, its corresponding feature is denoted as $z$. We select the threshold value as 0.5. The prediction $\hat{y}$ can be computed:

$$\hat{y} = \begin{cases} \text{Model A}(x) & \text{if } ||z||_2 < 0.5 \\ \text{Model B}(x) & \text{otherwise} \end{cases} \quad \text{(A.2)}$$

The results of Tab. 1 in the main text demonstrate that this ensemble-based approach is effective across most network architectures.

## A.3. Baseline Methods

We integrate S&D in representative methods to validate its effectiveness across image classification, object detection and semantic segmentation tasks.

**Robustness-related paradigms.** We compare S&D against four representative categories of robustness-enhancing methods: (1) Sub-network Selection, including AdaSAP (Bair et al., 2024) (results cited from the original paper) and Stochastic Depth (Huang et al., 2016) (reproduced following the original protocols); (2) Dynamic Weight Evolution, such as EWS (Guo et al., 2022), DST (Wu et al., 2025), and DAMP (Trinh et al., 2024), where results are cited from their respective publications; (3) Consistency Learning, featuring DAT (Mao et al., 2022) (evaluated using the official checkpoint) and TVM (Saikia et al., 2021) (reproduced following

original settings); and (4) Bio-inspired Designs, such as VOneNet (Dapello et al., 2020) and GaborNet (Pérez et al., 2020), both of which are strictly reproduced.

**Image classification.** We evaluate four aspects: (1) neural network backbones, including convolutional architectures such as AlexNet (Krizhevsky et al., 2012), MobileNetV2 (Sandler et al., 2018), SqueezeNet (Iandola, 2016), VGG (Simonyan, 2014), ResNet (He et al., 2016), ResNeXt (Xie et al., 2017), WideResNet (Zagoruyko, 2016), and ConvNeXt (Liu et al., 2022), as well as attention- and state-space-based models including Vision Transformer (ViT) (Dosovitskiy, 2020), MobileViT (Mehta & Rastegari, 2021), Efficientformer (Li et al., 2022), and MambaOut (Yu & Wang, 2025); detailed results for ViT and Mamba-based architectures are reported in Table 3; (2) data augmentation (e.g., AugMix (Hendrycks et al., 2019), AutoAug (Cubuk et al., 2018), RandAug (Cubuk et al., 2020), TriAug (Müller & Hutter, 2021), Random Erasing (Zhong et al., 2020)); (3) model regularization (e.g., CutMix (Yun et al., 2019), Mixup (Zhang et al., 2018), and Label smoothing (Müller et al., 2019)) and (4) a modern Pytorch training recipe (Paszke et al., 2019) combining diverse augmentations, regularization, and extended training schedules.

**Object detection&semantic segmentation.** We test both two-stage detectors (e.g., Faster RCNN (Ren et al., 2016), Mask RCNN (He et al., 2017), Cascade RCNN, and Cascade Mask RCNN (Cai & Vasconcelos, 2018))and one-stage detectors (RetinaNet (Ross & Dollár, 2017), YOLOv5 and YOLOv8 (Reis et al., 2023)). For semantic segmentation, we evaluate on multi-scale fusion-based (e.g., DeepLabV3+ (Chen et al., 2018), PSP-Net (Zhao et al., 2017)) and attention-based architectures (e.g., Mask2Former (Cheng et al., 2022), ANN (Zhu et al., 2019), CCNet (Huang et al., 2019), DANet (Fu et al., 2019), GCNet (Cao et al., 2019), and PSANet (Zhao et al., 2018)).

**Test-time adaptation methods.** To further evaluate the compatibility and effectiveness of S&D under distribution shifts, we integrate it with representative test-time adaptation (TTA) methods, including AdaContrast (Chen et al., 2022), BN statistics update (Benz et al., 2021), Tent (Wang et al., 2021), EATA (Niu et al., 2022), SANTA (Chakrabarty et al., 2023), SAR (Niu et al., 2023), RMT (Döbler et al., 2023) and RoTTA (Yuan et al., 2023), under the continual test-time adaptation setting. In this setting, models adapt sequentially to streaming corrupted data without access to the original training distribution. As shown in Tab. 5 of the main work, S&D consistently improves performance across various TTA methods and under diverse corruption types.

*Table A.1.* Hyperparameters of the baseline training recipe and advanced training recipe.

| | Baseline | Advanced | | Baseline | Advanced |
|---|---|---|---|---|---|
| Train Res | 224 | 176 | Label smoothing $\epsilon$ | ✗ | 0.1 |
| Test Res | 224 | 232 | Repeated Aug | ✗ | 4 |
| Epochs | 90 | 600 | H. flip | ✔ | ✔ |
| Batch size | 256 | 1024 | RRC | ✔ | ✔ |
| Optimizer | SGD-M | SGD-M | Tri Augment | ✗ | ✔ |
| LR | 0.1 | 0.5 | Mixup alpha | ✗ | 0.2 |
| LR Decay | step | cosine | Cutmix alpha | ✗ | 1.0 |
| decay rate | 0.1 | - | Erasing prob. | ✗ | 0.1 |
| decay epochs | 30 | - | ColorJitter | ✔ | ✔ |
| Weight decay | $10^{-4}$ | $2 \times 10^{-5}$ | EMA | ✗ | ✔ |
| Warmup Epochs | ✗ | 5 | CEloss | ✔ | ✔ |

## A.4. Training Configurations

**Neural network backbone.** Convolutional architectures are trained using the official PyTorch training recipes (Paszke et al., 2019). For the Vision Transformer (ViT), we follow the training protocol of DeiT (Touvron et al., 2021), trained from scratch without knowledge distillation. Other Transformer-based and Mamba-based models are trained from scratch according to the original procedures described in their respective papers, using implementations from the timm library (Wightman, 2019).

**Data augmentation and model regularization.** We consistently use ResNet-50 with the following configuration: 90 epochs, batch size of 256, initial learning rate of 0.1, SGD optimizer, and a StepLR scheduler that decays the learning rate by a factor of 0.1 every 30 epochs. All techniques are implemented using standard PyTorch modules, with hyperparameters listed in Table A.2.

*Table A.2.* Hyperparameter settings for data augmentation and model regularization strategies.

| AugMix | RandAug | Random Erasing |
|---|---|---|
| severity=3 | magnitude=9 | probability=0.1 |
| **CutMix** | **Mixup** | **Label Smoothing** |
| $\alpha$=0.1 | $\alpha$=0.1 | $\epsilon$=0.1 |

**Advanced training recipe.** We adopt a new PyTorch training protocol for validation and conduct experiments on MobileNetV2, ResNet-50, ResNeXt-50 and ConvNeXt. The comparison with the baseline is shown in Table A.1.

**Test-time adaptation method.** We evaluate performance under continual test-time adaptation on ImageNet-C at severity level 5, using the same open-source codebase (Döbler et al., 2024), with the batch size set to 64 for all methods.

## B. Ablation Studies and Analysis

### B.1. Ablation of MDSM and SPTS

We conduct ablation studies on the two core components of S&D. As shown in Table B.1, applying only MDSM slightly improves ImageNet-C robustness (33.5 vs. 32.9), while combining MDSM with SPTS yields the best corruption robustness (34.8), with minimal impact on clean accuracy.

*Table B.1.* Ablation results of S&D using ResNet-18 on ImageNet and ImageNet-C. Results are reported in top-1 accuracy (%), with the highest values indicated in **bold**.

| MDSM | SPTS | ImageNet | ImageNet-C |
|---|---|---|---|
| ✗ | ✗ | 69.2 | 32.9 |
| ✔ | ✗ | **69.3** | 33.5 |
| ✔ | ✔ | 68.6 | **34.8** |

### B.2. Choice of Tweaking Operations

In SPTS, we adopt GeoDiversify, a geometric diversification strategy based on permutation and spatial transformations, to enhance robust feature diversity while preserving structural consistency. See Section A.1 for details. We also explore several other structure-consistent tweaking operations, including **WeightMix**, which performs weighted mixing between robust and non-robust pathways; **GaussPerturb**, which adds Gaussian noise to the weights of robust pathways; **FreqPerturb**, which applies perturbations in the frequency domain; **LowRankPerturb**, which applies perturbations along the smallest singular vector after SVD; and **UniPerturb**, which injects uniform noise into robust pathway weights. Results in Table B.2 show that **GeoDiversify** achieves the best trade-off between robustness gain and structural fidelity, and thus we use it as the default operation in our main experiments.

### B.3. Choice of Corruption-generating Transforms

As mentioned in Sec. 3.3.2, we synthesize corrupted inputs $\hat{X}$ from clean images $X$ using simple transformations to

*Table B.2.* The results of different tweaking operations using ResNet-18 for ImageNet, ImageNet-C, and ImageNetV2-C. The highest accuracy is indicated in **bold**.

|  | ImageNet | ImageNet-C | ImageNetV2-C |
|---|---|---|---|
| Baseline | **69.2** | 32.9 | 24.4 |
| WeightMix | 69.1 | 34.9 | 26.3 |
| GaussPerturb | 68.6 | 35.3 | 26.6 |
| UniPerturb | 68.5 | 35.5 | 26.7 |
| FreqPerturb | 68.6 | 35.5 | **26.8** |
| LowRankPerturb | 68.4 | 35.3 | 26.6 |
| GeoDiversify | 68.4 | **35.6** | **26.8** |

guide robust pathway selection in MDSM. We evaluate several corruption strategies: **Contrast+Noise** (Saikia et al., 2021), combining contrast adjustment and additive noise; **ColorJitter**, randomly perturbing brightness, contrast, saturation, and hue; **LowFreq**, adding structured low-frequency perturbations; **MixedFreq**, injecting noise in both low- and high-frequency components; **Solarize**, randomly inverting pixels above a threshold; **AutoAugment** (AutoAug), applying ImageNet-optimized policies; and **RandAugment** (RandAug), using randomized augmentations with uniform magnitude. Results in Table B.3 show that while all methods provide useful supervision for MDSM, **Contrast+Noise** achieves the best trade-off between simplicity and effectiveness, and is thus adopted in our main experiments.

*Table B.3.* The results of ResNet-18 under different corruption-generating transforms on ImageNet, ImageNet-C, and ImageNetV2-C. Results are reported in top-1 accuracy (%), with the highest values indicated in **bold**.

|  | ImageNet | ImageNet-C | ImageNetV2-C |
|---|---|---|---|
| Baseline | **69.2** | 32.9 | 24.4 |
| ColorJitter | 68.8 | 33.2 | 24.9 |
| LowFreq | 68.0 | 34.8 | 26.2 |
| MixedFreq | 67.8 | **34.9** | 26.2 |
| Solarize | 68.8 | 33.8 | 25.4 |
| AutoAug | 68.8 | 34.2 | 25.6 |
| RandAug | 68.5 | 34.7 | 26.0 |
| Contrast+Noise | 68.4 | **34.9** | **26.3** |

### B.4. Choice of Similarity Metrics

We evaluate the effectiveness of various similarity metrics used in Eq. (6), including the $l_1$ norm, $l_2$ norm, SVCCA (Raghu et al., 2017), and CKA (Kornblith et al., 2019). As shown in Table B.4, all metrics improve corruption robustness compared to the baseline, demonstrating the flexibility of our method with respect to the choice of similarity measure. While $l_1$ and $l_2$ norms yield slightly higher robustness on ImageNet-C, they also lead to a more significant drop in clean accuracy on ImageNet. In contrast, CKA achieves a better trade-off between robustness and clean performance. Therefore, we adopt CKA as the similarity metric in our main experiments to balance accuracy

on both clean and corrupted data.

*Table B.4.* The results of different similarity metrics using ResNet-18 for ImageNet, ImageNet-C, and ImageNetV2-C. Results are reported in top-1 accuracy (%), with the highest values indicated in **bold**.

|  | ImageNet | ImageNet-C | ImageNetV2-C |
|---|---|---|---|
| Baseline | 69.2 | 32.9 | 24.4 |
| $l_1$ | 67.4 | **36.8** | **27.8** |
| $l_2$ | 67.3 | 36.4 | 27.6 |
| SVCCA | **69.6** | 33.2 | 24.8 |
| CKA | 68.6 | 34.8 | 26.2 |

### B.5. Choice of TopK

We conduct an ablation study on the TopK selection ratio in Eq. (7), evaluating settings of 5%, 10%, 15%, 30%, 40%, and 50%. As shown in Table B.5, selecting TopK in the range of 30% to 50% yields a better trade-off between clean accuracy and corruption robustness, with 50% (i.e., retaining half the pathways) serving as a balanced and effective default in our framework.

*Table B.5.* The results of different settings on Top-$K$ using ResNet-18 for ImageNet, ImageNet-C and ImageNetV2-C. Results are reported in top-1 accuracy (%), with the highest values indicated in **bold**.

|  | ImageNet | ImageNet-C | ImageNetV2-C |
|---|---|---|---|
| Baseline | **69.2** | 32.9 | 24.4 |
| 5% | 59.0 | 28.2 | 21.3 |
| 10% | 62.4 | 31.1 | 23.2 |
| 15% | 66.0 | 34.6 | 26.1 |
| 30% | 68.4 | 35.3 | 26.6 |
| 40% | 68.6 | **35.4** | **26.7** |
| 50% | 69.1 | 34.3 | 25.9 |

### B.6. Position for Applying S&D

As highlighted in Sec. 3 of our main work, the S&D refinement can be flexibly applied to any sub-network of a deep model. To investigate how its placement affects performance, we integrate S&D at different depths of ResNet-18, specifically after the sub-networks defined by $f_{l\leq1}$, $f_{l\leq3}$, $f_{l\leq5}$, $f_{l\leq7}$, and $f_{l\leq9}$. We evaluate these variants on ImageNet, ImageNet-C, and ImageNetV2-C, with results reported in Table B.6. The findings show that applying S&D at the shallowest position ($f_{l\leq1}$) yields the highest corruption robustness, while positions $f_{l\leq3}$ and $f_{l\leq5}$ still provide meaningful improvements. However, as S&D is placed deeper in the network, such as at $f_{l\leq7}$ or $f_{l\leq9}$, the robustness gain gradually diminishes. This trend demonstrates that refining pathways early, before non-robust features propagate through subsequent layers, is essential for achieving optimal robustness.

*Table B.6.* Accuracy of ResNet-18 when applying S&D at varying sub-network depths on ImageNet, ImageNet-C, and ImageNetV2-C. Results are reported in top-1 accuracy (%), with the highest values indicated in **bold**.

|  | ImageNet | ImageNet-C | ImageNetV2-C |
|---|---|---|---|
| Baseline | **69.2** | 32.9 | 24.4 |
| $f_{l \leq 1}$ | 68.6 | **34.8** | **26.2** |
| $f_{l \leq 3}$ | 69.1 | 33.2 | 24.7 |
| $f_{l \leq 5}$ | 68.6 | 33.0 | 24.8 |
| $f_{l \leq 7}$ | 67.7 | 32.1 | 24.0 |
| $f_{l \leq 9}$ | 66.5 | 30.6 | 22.7 |

## B.7. Sensitivity to Random Seeds

To assess the stability of S&D, we repeat the experiments across multiple random seeds. The results, summarized in Table B.7, show consistent improvements in corruption robustness regardless of the seed, with minimal variance in both clean and corrupted accuracy. This indicates that the performance gains introduced by S&D are stable and not attributable to favorable random initialization.

*Table B.7.* The results of different random seeds using ResNet-18+S&D for ImageNet, ImageNet-C, and ImageNetV2-C. # denotes the seed index. Results are reported in top-1 accuracy (%), with the highest values indicated in **bold**.

|  | ImageNet | ImageNet-C | ImageNetV2-C |
|---|---|---|---|
| Baseline | **69.2** | 32.9 | 24.4 |
| #1 | 68.5 | 34.2 | 25.8 |
| #2 | 68.4 | 35.1 | 26.5 |
| #3 | 68.2 | 34.9 | 26.3 |
| #4 | 68.5 | 35.3 | 26.6 |
| #5 | 68.3 | 35.4 | 26.6 |
| #6 | 68.2 | 35.0 | 26.4 |
| #7 | 68.2 | 35.8 | 27.0 |
| #8 | 68.5 | **37.6** | **28.9** |

## B.8. Sensitivity to Learning Rate

As discussed in Sec. 3.3.3 of our main work, the optimal pathway group $G^*$ produced by S&D is integrated into the neural network for standard training. Here, we evaluate different learning rates applied to $G^*$ during this phase. Results in Table B.8 show that, compared to the baseline, all tested learning rates improve corruption robustness, albeit with a slight drop in clean accuracy. In our main experiments, we fix the learning rate for $G^*$ at 0.0 to preserve the refined robust features without further adaptation.

## C. Extended Experimental Results

### C.1. Extended Results on Image Classification

We present additional image classification results in Table D.1 and Table 3. Table D.1 reports performance on corruption benchmarks using various convolutional neural networks not included in the main text, showing that S&D

*Table B.8.* The results of different learning rates using ResNet-18+S&D for ImageNet, ImageNet-C, and ImageNetV2-C. Results are reported in top-1 accuracy (%), with the highest values indicated in **bold**.

|  | ImageNet | ImageNet-C | ImageNetV2-C |
|---|---|---|---|
| Baseline | **69.2** | 32.9 | 24.4 |
| 0.0 | 68.6 | 34.8 | **26.2** |
| 0.001 | 67.6 | **35.0** | **26.2** |
| 0.01 | 67.9 | 34.3 | 25.6 |
| 0.1 | 68.4 | 34.4 | 25.6 |
| 1 | 68.8 | 34.5 | 25.6 |

consistently improves robustness across various settings.

### C.2. Fine-Grained Corruption Robustness

We provide a per-corruption breakdown of ImageNet-C performance for MobileNetV2, ResNet-50, and ConvNeXt-base in Table D.2. This complements the averaged robustness results and highlights how S&D performs across individual corruption types. S&D consistently improves robustness across most corruptions: all 15/15 corruption types on MobileNetV2, 12/15 on ResNet-50, and 13/15 on ConvNeXt-base, totaling 40 out of 45 architecture-corruption pairs. Gains span noise, blur, weather, and digital corruptions, confirming the method is not overfitting to specific corruptions. Some exceptions exist, such as Snow, Fog, and Brightness on ResNet-50, and Elastic and Pixelate on ConvNeXt-base, indicating areas for further investigation. Overall, this analysis demonstrates that S&D provides broad, architecture-agnostic improvements in corruption robustness, while also identifying specific corruptions where improvements are limited.

## D. Additional Visualizations

### D.1. Extended Visualizations of Robust and Non-robust Features

In Sec. 3.2.2, we present the cumulative distributions of $\gamma$-robust features across sub-networks under two common corruptions, including Gaussian Noise and Fog. Here, we extend this analysis by providing visualization results for an additional 13 corruption types, as shown in Figure D.1. Notably, the patterns observed under these 13 corruptions align closely with those reported in the main work, further corroborating our findings.

### D.2. Fine-grained Analysis of Robust vs. Non-robust Feature Pathways

In Sec 3.2.3, we analyze the impact of selectively retaining $\gamma$-robust versus non-$\gamma$-robust pathways on model robustness under image corruptions, as summarized in Fig. 3 of the main work. Here, we provide extended results for this anal-

ysis across all 15 corruption types, as shown in Figure D.2. The consistent advantage of $\gamma$-robust pathways in enhancing effective robustness is evident across all corruption types, further reinforcing the critical role of robust feature aggregation in the robustness of downstream sub-networks and the full model.

### D.3. The Visualization on Semantic Segmentation

We further test S&D on the ACDC (Sakaridis et al., 2021) dataset, a semantic segmentation under real-world adverse weather conditions, including Fog, Rain, Night, and Snow. The visualization of segmentation results using GCNet (Cao et al., 2019) (with and without S&D) is shown in Figure D.3. S&D reduces the segmentation errors across different adverse weather scenarios.

*Table D.1.* We report Top-1 Accuracy (ACC) for ImageNet and mean Corruption Error (mCE) for its corrupted variants (C, $\bar{C}$, 3DCC, and V2-C). Avg.mCE averages these four error rates. [·] indicates ensemble-based results; (+) and (−) denote performance changes via S&D. * indicates advanced training recipes.

| Main | S&D | ImageNet | ImageNet-C | ImageNet-$\bar{C}$ | ImageNet-3DCC | ImageNetV2-C | Avg.mCE (↓) |
|---|---|---|---|---|---|---|---|
| | | | Backbones of Neural Network | | | | |
| AlexNet | ✗ | 55.6 | 100.0 | 100.0 | 100.0 | 100.0 | 100.0 |
| | ✔ | 55.1 [55.4] | 98.2 | 99.8 | 99.4 | 98.5 | 99.0 (**-1.0**) |
| SqueezeNet1_0 | ✗ | 57.1 | 103.5 | 101.6 | 100.4 | 102.2 | 101.9 |
| | ✔ | 57.9 [57.3] | 100.1 | 99.7 | 98.1 | 99.6 | 99.4 (**-1.5**) |
| SqueezeNet1_1 | ✗ | 57.3 | 104.4 | 101.5 | 100.3 | 102.7 | 102.2 |
| | ✔ | 58.3 [57.9] | 102.2 | 100.7 | 99.0 | 101.1 | 100.8 (**-1.4**) |
| ResNet-18 | ✗ | 69.2 | 84.7 | 87.0 | 82.1 | 88.9 | 85.7 |
| | ✔ | 68.6 [69.2] | 82.4 | 86.1 | 81.4 | 86.8 | 84.2 (**-1.5**) |
| ResNet-34 | ✗ | 72.8 | 77.9 | 81.4 | 76.3 | 83.8 | 79.9 |
| | ✔ | 71.9 [72.8] | 74.6 | 80.1 | 74.7 | 81.0 | 77.6 (**-2.3**) |
| ResNet-101 | ✗ | 77.0 | 70.4 | 74.1 | 68.6 | 77.8 | 72.7 |
| | ✔ | 76.0 [77.0] | 66.7 | 74.3 | 67.0 | 74.7 | 70.7 (**-2.0**) |
| VGG16 | ✗ | 73.3 | 84.6 | 85.8 | 80.7 | 88.7 | 85.0 |
| | ✔ | 75.1 [73.9] | 82.9 | 84.0 | 79.2 | 87.5 | 83.4 (**-1.6**) |
| ResNeXt-50 | ✗ | 77.4 | 72.3 | 74.6 | 69.8 | 79.6 | 74.1 |
| | ✔ | 76.1 [77.4] | 70.8 | 75.3 | 70.0 | 77.9 | 73.5 (**-0.6**) |
| ResNeXt-101 | ✗ | 79.1 | 66.7 | 69.4 | 65.1 | 74.9 | 69.0 |
| | ✔ | 77.4 [79.1] | 63.0 | 70.2 | 64.2 | 71.7 | 67.3 (**-1.7**) |
| WideResNet-101_2 | ✗ | 78.8 | 67.7 | 71.8 | 66.1 | 76.2 | 70.5 |
| | ✔ | 77.1 [77.8] | 65.7 | 72.0 | 65.7 | 74.0 | 69.4 (**-1.1**) |
| | | | Data Augmentation & Model Regularization | | | | |
| Random Erasing | ✗ | 76.4 | 76.6 | 78.3 | 72.4 | 82.4 | 77.4 |
| | ✔ | 74.8 [76.4] | 72.9 | 78.3 | 71.6 | 79.6 | 75.6 (**-1.8**) |
| AutoAug | ✗ | 76.4 | 73.2 | 76.9 | 69.6 | 79.8 | 74.9 |
| | ✔ | 74.9 [76.4] | 68.6 | 76.7 | 67.9 | 76.1 | 72.3 (**-2.6**) |
| RandAug | ✗ | 76.5 | 73.7 | 75.6 | 70.3 | 80.1 | 74.9 |
| | ✔ | 75.1 [76.3] | 68.8 | 76.5 | 68.1 | 76.4 | 72.5 (**-2.4**) |
| CutMix | ✗ | 76.9 | 76.9 | 76.4 | 72.5 | 82.5 | 77.1 |
| | ✔ | 75.1 [76.9] | 72.8 | 76.4 | 71.0 | 79.2 | 74.9 (**-2.2**) |
| | | | Advanced Training Recipe | | | | |
| ResNeXt-50* | ✗ | 81.1 | 62.6 | 62.4 | 59.8 | 71.3 | 64.0 |
| | ✔ | 78.7 [79.0] | 60.1 | 63.5 | 61.5 | 69.8 | 63.7 (**-0.3**) |
| ConvNeXt-tiny* | ✗ | 82.5 | 60.9 | 58.3 | 59.8 | 69.5 | 62.1 |
| | ✔ | 81.8 [82.1] | 57.0 | 57.0 | 57.8 | 66.7 | 59.6 (**-2.5**) |
| ConvNeXt-small* | ✗ | 83.5 | 56.2 | 55.7 | 56.0 | 65.8 | 58.4 |
| | ✔ | 82.6 [82.9] | 53.5 | 54.0 | 55.0 | 63.8 | 56.6 (**-1.8**) |

*Table D.2.* We report Top-1 Accuracy (ACC) for ImageNet and mean Corruption Error (mCE) for its corrupted variants (ImageNet-C with 15 corruption types). Avg.mCE averages these 15 error rates. [·] indicates ensemble-based results; (+) and (−) denote performance changes via S&D. * indicates advanced training recipes.

| Main | S&D | ImageNet | Gauss. | Shot | Impulse | Defocus | Glass | Motion | Zoom | Snow | Frost | Fog | Bright | Contrast | Elastic | Pixel | JPEG |
|---|---|---|---|---|---|---|---|---|---|---|---|---|---|---|---|---|---|
| MobileNetV2* | ✗ | 72.1 | 89.7 | 89.9 | 86.5 | 87.4 | 96.9 | 86.7 | 88.5 | 82.3 | 83.3 | 67.8 | 65.7 | 64.7 | 93.8 | 115.6 | 92.6 |
| | ✔ | 72.5 [72.3] | 76.8 | 77.2 | 78.2 | 82.5 | 92.7 | 81.7 | 87.1 | 80.6 | 75.3 | 65.9 | 65.0 | 61.4 | 90.8 | 81.1 | 81.2 |
| ResNet-50* | ✗ | 80.7 | 67.5 | 68.3 | 68.7 | 67.5 | 85.9 | 73.8 | 72.0 | 64.6 | 62.0 | 48.8 | 46.4 | 46.3 | 79.5 | 80.7 | 65.4 |
| | ✔ | 79.5 [79.7] | 57.0 | 57.1 | 57.4 | 65.1 | 78.5 | 69.0 | 71.2 | 67.0 | 58.2 | 52.5 | 50.2 | 45.9 | 75.5 | 54.7 | 63.7 |
| ConvNeXt-base* | ✗ | 84.0 | 50.0 | 51.6 | 49.6 | 60.1 | 72.6 | 53.6 | 63.3 | 50.2 | 46.8 | 50.8 | 39.4 | 39.4 | 63.0 | 58.6 | 54.6 |
| | ✔ | 82.9 [83.4] | 44.8 | 45.0 | 44.2 | 57.4 | 69.0 | 51.9 | 60.6 | 47.0 | 43.1 | 44.3 | 38.4 | 37.0 | 64.6 | 60.1 | 53.8 |

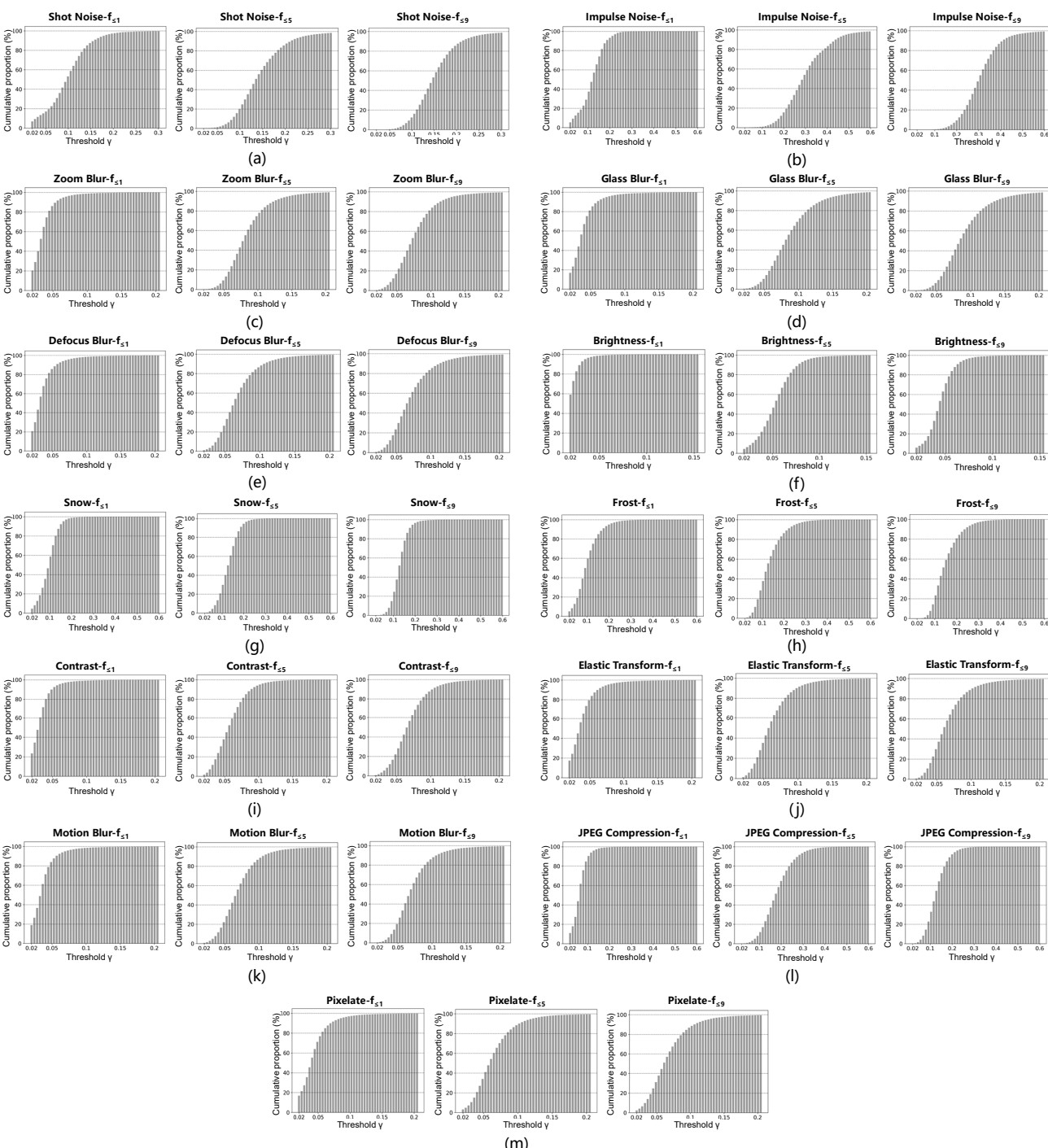

*Figure D.1.* The cumulative distribution of γ-robust features at different sub-networks under the corruption of (a) Shot Noise, (b) Impulse Noise, (c) Zoom Blur, (d) Glass Blur, (e) Defocus Blur, (f) Brightness, (g) Snow, (h) Frost, (i) Contrast, (j) Elastic Transform and (k) Pixelate.

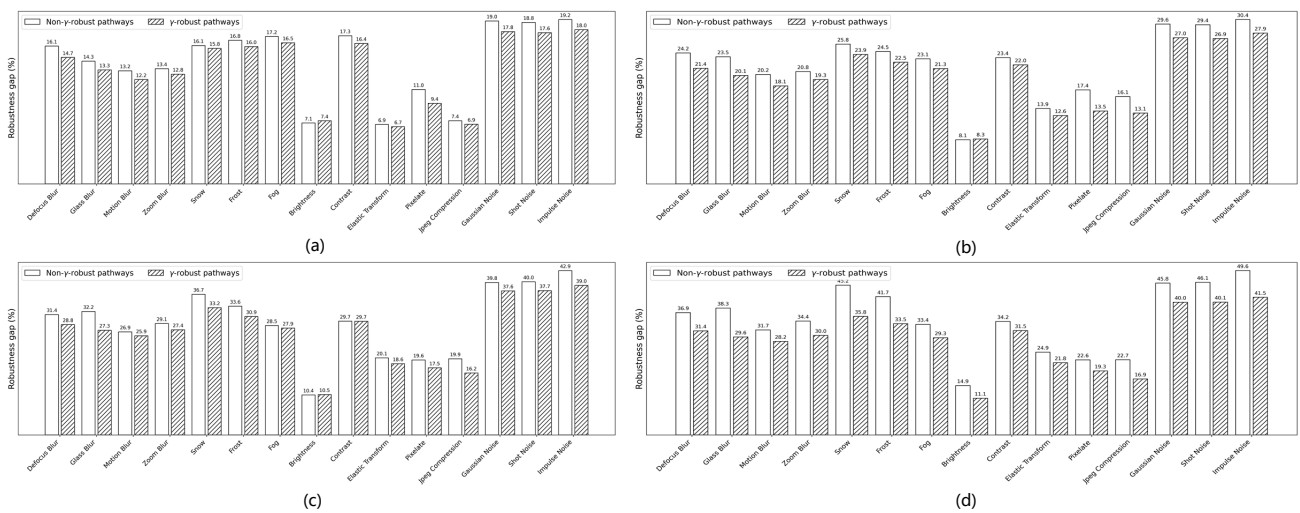

*Figure D.2.* Effective robustness of sub-networks (a) $f_{\leq 1}$, (b) $f_{\leq 5}$, (c) $f_{\leq 9}$, and (d) the full model $f_\theta$ when preserving either $\gamma$-robust or non-$\gamma$-robust pathways across 15 corruption types. Lower effective robustness indicates better performance.

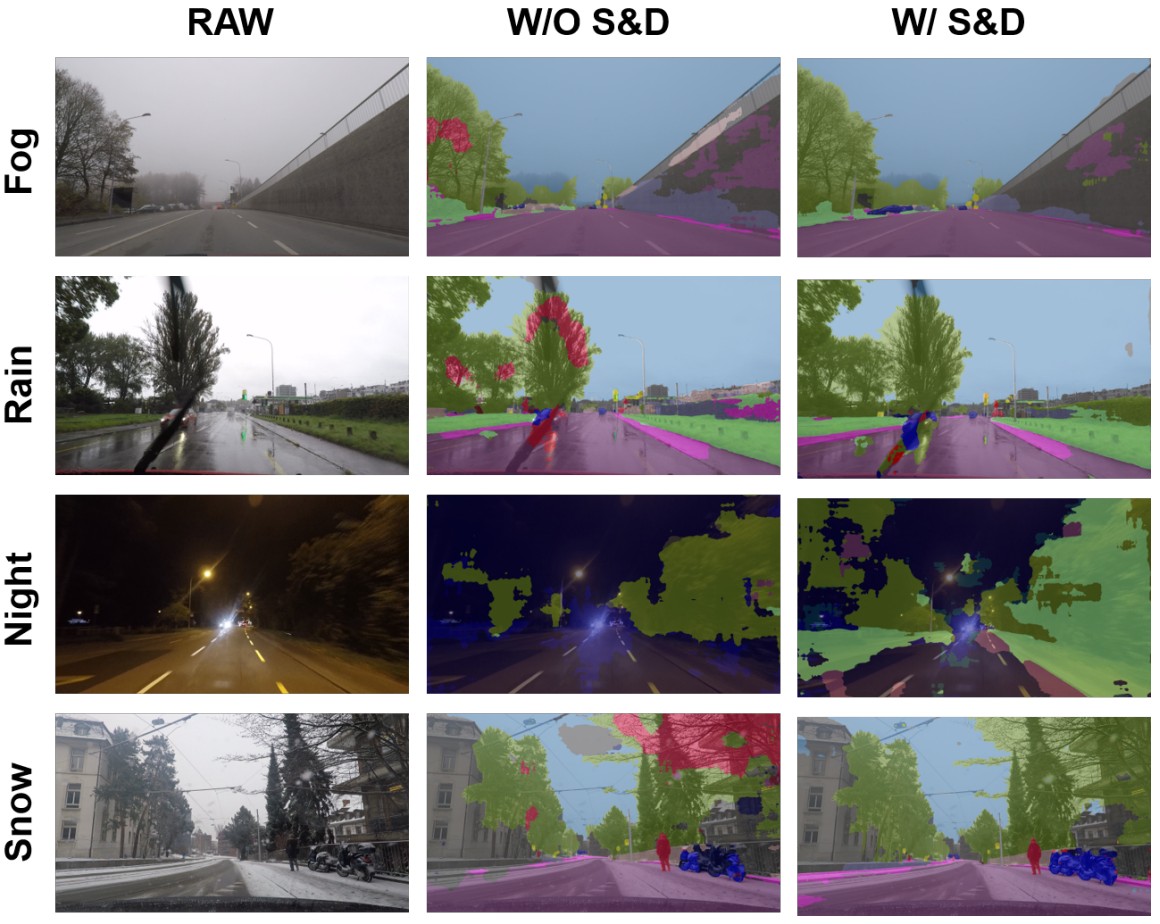

*Figure D.3.* The visualization of segmentation results for the ACDC dataset.

