# OpenReview forum: "Suppress and Diversify: Refining Robust Pathways for Corruption Robustness"
_ICML.cc/2026/Conference — ICML 2026 spotlight_

### Official Review · Reviewer_Vx6W · 2026-03-10

**Soundness:** 3
**Presentation:** 3
**Significance:** 3
**Originality:** 3
**Overall Recommendation:** 4
**Confidence:** 2

**Summary:**

The paper proposes Suppress and Diversify (S&D), a training-time mechanism aimed at improving corruption robustness by manipulating internal computational pathways.
S&D has two components: (1) MDSM, which maintains a pathway-group memory bank, iteratively comparing and updating the stored group against candidate groups using a fitness score computed from clean images and their synthetically corrupted counterparts. (2) SPTS, which diversifies the robust pathway group by first TopK-filtering robust pathways and then generating additional tweaked robust pathways via symmetry-preserving transformations of their weights to form an augmented group for training.
The method is architecture-agnostic and parameter-free, requiring no structural modifications and incurring zero test-time overhead.
The manuscript contains empirical analysis of feature robustness in standard neural networks, and extensive experiments evaluating the proposed method across image classification, detection, and segmentation robustness benchmarks.

**Compliance With Llm Reviewing Policy:**

Affirmed.

**Final Justification:**

The rebuttal addressed my concerns. I retain my original positive score.

**Key Questions For Authors:**

1. Can the authors report the absolute clean and corrupted accuracies (not only the robustness gap) for the robust-only vs non-robust-only pathway variants used in Remark 2’s analysis?

2. Can the authors precisely specify how $G^t$ is constructed, what an “iteration” nnn corresponds to in Eq. (6), and how often MDSM is run relative to training epochs?

3. Can the authors add a brief discussion that explicitly summarizes the clean-accuracy vs corruption-robustness tradeoff across cases where it exists?

4. Can the authors report the training-time overhead of S&D (e.g., wall-clock runtime and memory consumption) relative to baseline training?

5. The transformer/state-space results (e.g., ViT/Mamba) appear to be reported on smaller-scale datasets (e.g., ImageNet-100). Can the authors either (i) report at least one full ImageNet-scale experiment for these architectures, or (ii) justify why the smaller-scale evaluation was used for these models?

**Limitations:**

The paper does not provide an explicit limitations and societal impact discussion.
The authors should (1) explicitly discuss the clean-accuracy vs robustness tradeoff, (2) report training-time overhead (wall-clock/compute/memory), (3) clarify a few algorithm details.

**Strengths And Weaknesses:**

**Strength**


1. The paper presents empirically grounded motivation by first providing evidence that (i) robust and non-robust features coexist, (ii) the proportion of robust features decays with depth across corruption types, and (iii) robust-feature prevalence is linked to robustness of sub-networks and the full model via robustness-gap and loss-landscape analyses. This makes the proposed pathway manipulation feel well-motivated rather than purely heuristic (with some caveat mentioned in weaknesses).

2. The proposed method is relatively simple and generic. It does not introduce extra learned modules (parameter-free), does not require architectural changes, and incurs no test-time overhead. The method is evaluated across multiple backbones (including CNN and transformer/state-space variants) and is shown to be compatible with common training recipes and robustness pipelines, reinforcing the “plug-and-play / architecture-agnostic” positioning.


3. The manuscript includes extensive empirical results across 3 domains (classification, detection, and segmentation), as well as ablations justifying design choices. The empirical evaluation shows consistent corruption-robustness gains across tasks and backbones. The paper also includes a fairly comprehensive ablation study (e.g., corruption generator for $\hat{x}$, similarity metric in MDSM, TopK ratio, tweaking operations in SPTS, insertion depth, and random-seed stability).

**Weaknesses**

1. The experiment leading to Remark 2 could be misleading since it reports only the robustness gap and does not account for the absolute accuracy of each model. For example, a trivial classifier could have a near-zero gap while still being useless. I suggest the authors report clean accuracy and corrupted accuracy (not only the gap) for each ablated model in this experiment.


2. There are several unclearities in the algorithm presentation.
    1. Why use the term “pathway group” (group vs set)? The method appears to treat it as an unordered collection, but this is not explicitly clarified.
    2. How is the candidate group $G^t$ selected/constructed in each update (e.g., deterministic from the current network, random subset, class-conditional, etc.)?
    3. The notation for the update loop is confusing: $t$ is used as an epoch index, while $n$ and $N$ are defined as “current and total iterations,” but it remains unclear what an “iteration” concretely is (e.g. epoch / minibatch), how $n$ progresses relative to epochs $t$, and how often MDSM is invoked during training.


3. Missing discussion of the accuracy–robustness tradeoff. In several ImageNet settings, clean accuracy decreases while corruption robustness improves; while some of this tradeoff appears in appendix ablations (e.g., similarity-metric choices), it is not discussed prominently in the main text.

4. Missing computational / training-cost analysis. The paper emphasizes no test-time overhead, but does not report training-time overhead (e.g., extra forward passes on $x,\, \hat{x}$, similarity computation, memory-bank updates), which is important for assessing practicality.

---

> ### Author Rebuttal · Authors · 2026-03-31
>
> We thank the reviewer for the careful reading and constructive feedback. The additional tables and figures referenced below are included in the anonymous supplementary material: **[Tables and Figures](https://anonymous.4open.science/r/sd_1/README.md)**.
>
> **Clarifying the Role of Robustness Gap in Remark 2.** Remark 2 is not intended to show that $\gamma$-robust-only pathways achieve better absolute accuracy, but to isolate their causal role in corruption robustness under the same probing protocol. As shown in Table S6 in the anonymous supplementary file, the $\gamma$-robust group consistently yields a smaller robustness gap across all subnetworks and the full model, while still maintaining non-trivial clean and corrupted accuracies, rather than collapsing to a trivial classifier. Since robustness gap is widely used in prior work to measure corruption sensitivity, we use it here as a robustness-oriented diagnostic. This conclusion is further supported by the consistent per-corruption analysis and the flatter loss landscapes of $\gamma$-robust pathways.
>
>  **Algorithm and Training Procedure Clarification.** We will clarify that the “pathway group” is not an unordered set, but a pathway collection with fixed indexing inherited from layer $l$, and revise the notation accordingly. We will also state explicitly that the candidate group $G^t$ is deterministically constructed from the current model state, rather than randomly sampled or class-conditional. For Eq. (6), $n$ and $N$ denote the current and total scheduled MDSM steps, respectively, both defined at the epoch level rather than the mini-batch level; thus, $n$ progresses with the epoch index when MDSM is invoked. MDSM is performed once between consecutive epochs when scheduled, and in the default setting runs during the first five epochs of training.
>
> **Clean Accuracy vs. Corruption Robustness Tradeoff.** In several ImageNet settings, S&D improves corruption robustness at the cost of some clean accuracy, and this tradeoff is not sufficiently discussed in the current main text. We view this as a characteristic of robustness-oriented pathway selection: S&D favors features that remain stable under corruption, which may suppress some brittle yet clean-accuracy-helpful features, consistent with prior robust/non-robust feature perspectives [1]. Although we do not claim a complete mechanistic explanation, this behavior should be made explicit. In the revision, we will briefly summarize where this tradeoff appears, discuss it as a limitation when clean accuracy is the primary objective, and clarify that Appendix A.2 studies an ensemble-based mitigation that helps recover clean accuracy while preserving robustness benefits. This mitigation is practical but not a complete solution, and more principled strategies remain future work.
>
> **Training-Time Overhead and Practicality.** We will add this analysis in the revision. As shown in Table S7 in the anonymous supplementary file, baseline training takes 2294.27 s/epoch, while MDSM adds only 0.3233 s on selected epochs (~0.01%), and the one-time SPTS cost is 0.0057 s (<0.001%). Within MDSM, CKA is the dominant cost (0.2757 s), while corrupted-input generation (0.0056 s) and forward propagation (0.0420 s) are minor. Crucially, MDSM's inter-epoch execution ensures it never competes for active GPU memory with the main training process. As shown in Table S8, MDSM also stays within the baseline memory budget (7640.79 MB vs. 21292.07 MB) and its memory bank is negligible, since it stores only the selected subnetwork weights (e.g., only tens of KB for the ResNet-50 stem). Overall, the added training overhead of S&D is negligible in both wall-clock time and memory.
>
> **Evaluation Scope for ViT and Mamba Models.** We acknowledge that full ImageNet-scale results would provide stronger evidence for transformer/state-space architectures. Our current use of ImageNet-100 serves as an efficient, controlled first-pass validation to explore whether the effect of S&D extends beyond CNNs, rather than as a definitive benchmark for ViT/Mamba. In the current submission, the primary large-scale evidence comes from ImageNet-1K results on CNN backbones, while the smaller-scale ViT/Mamba experiments offer complementary cross-architecture evidence. We will clarify this scope in the revision and extend evaluations to larger scales in future work to avoid overgeneralization.
>
> **Limitations and Broader Discussion.** We will add a dedicated discussion in the revision to: (1) quantify the clean-accuracy vs. robustness tradeoff; (2) detail the minimal practical overhead and algorithmic specifics; and (3) discuss broader impacts, noting that while S&D enhances reliability in safety-critical deployments, it should be integrated with broader considerations of efficiency and dataset bias.
>
> [1] Ilyas A, Santurkar S, Tsipras D, et al. Adversarial examples are not bugs, they are features[J]. Advances in neural information processing systems, 2019, 32.

---

> > ### Author Rebuttal · Reviewer_Vx6W · 2026-04-02
> >
> > I thank the authors for the rebuttal. I appreciate their engagement with my questions, the added result details, and the additional overhead analysis.
> > While most of my concerns were addressed, I would like to note two remaining suggestions.
> > First, Table S6 is helpful in showing that the \gamma-robust pathways do not collapse to a trivial classifier. At the same time, I think it may be worth explicitly noting as a limitation or discussion point that, under this probing setup, the non-\gamma-robust pathways still form better classifiers (in terms of both clean and corrupted accuracy), even though the \gamma-robust pathways have the smaller robustness gap. While I understand that “robustness gap” may be widely used in the relevant literature, I think briefly clarifying this nuance is important and also connects naturally to the broader accuracy-robustness tradeoff.
> > Second, I would still encourage the authors to explain the notion of “pathway group” more explicitly in the revision. As currently written, it remains somewhat unclear whether this should be understood as a formal mathematical object (“group”) or simply as a collection of pathways from the selected layer. A more concrete description of how the candidate group is constructed would make the method easier to follow.
> > Overall, I believe incorporating the rebuttal material would improve the paper, and I retain my original positive score.

---

> > > ### Author Response · Authors · 2026-04-03
> > >
> > > We thank the reviewer for the constructive follow-up and for the positive assessment.
> > >
> > > Regarding Table S6, we agree that this nuance should be stated more explicitly in the revision. Under the probing setup of Remark 2, the non-γ-robust pathways indeed achieve higher absolute clean and corrupted accuracy, whereas the γ-robust pathways consistently exhibit a smaller robustness gap. We therefore do not intend Remark 2 to suggest that γ-robust pathways form stronger classifiers in absolute terms. Rather, its purpose is to isolate their role in reducing corruption sensitivity under the same probing protocol. We agree that this distinction is important, and we will add it explicitly to the discussion, together with a clearer connection to the broader clean-accuracy versus corruption-robustness tradeoff.
> > >
> > > We also thank the reviewer for pointing out the terminology issue around “pathway group.” In our method, this term is not used in the algebraic sense. Instead, it refers to an indexed collection of pathways associated with the subnetwork before the S&D insertion point. We use this formulation because S&D operates on these pathways as a structured unit: the pathways in the collection are jointly scored and selected in MDSM, and the selected robust pathways are further transformed in SPTS. More concretely, the candidate collection $G^t$ is deterministically constructed from the current model state at each scheduled MDSM epoch, with fixed indexing inherited from layer $l$, rather than by random or class-conditional sampling. We will revise the paper to make this construction and interpretation explicit, so that the algorithm is easier to follow.
> > >
> > > We appreciate these suggestions and agree that incorporating both clarifications will improve the paper.

---

### Official Review · Reviewer_tW1X · 2026-03-13

**Soundness:** 3
**Presentation:** 3
**Significance:** 3
**Originality:** 3
**Overall Recommendation:** 5
**Confidence:** 3

**Summary:**

This paper studies corruption robustness through the lens of internal computational pathways. The central observation is that, under natural corruptions, robust features undergo progressive decay across network depth, and that the prevalence of robust pathways is closely tied to the robustness of both downstream sub-networks and the model as a whole. Building on this analysis, the paper proposes Suppress and Diversify (S&D), a lightweight training-time refinement comprising two components: a Memory-aware Dynamic Selection Mechanism (MDSM) for identifying robust pathways, and a Structure-consistent Path Tweaking Strategy (SPTS) for diversifying them via symmetry-preserving transformations. The method is non-intrusive, architecture-agnostic, and adds no overhead at test time. It is evaluated across classification, detection, and segmentation tasks, as well as in compatibility settings with existing robustness methods.

**Compliance With Llm Reviewing Policy:**

Affirmed.

**Final Justification:**

I appreciate the rebuttal provided by the authors. Most of my concerns have been addressed. I suggest including the training-time measurement table and the Fine-Grained Robustness experiments in the revised manuscript or supplementary material.

I will raise my recommendation score.

**Key Questions For Authors:**

Please refer to **Strengths And Weaknesses**.

**Limitations:**

Yes

**Strengths And Weaknesses:**

**Strengths**

1. The S&D framework is conceptually clear and practically appealing. MDSM selects robust pathways by jointly considering invariance and discriminability, while SPTS diversifies them through structure-consistent transformations to prevent over-specialization. The method requires no architectural modification and introduces no test-time overhead, making it attractive from both research and deployment perspectives.
2. The evaluation is comprehensive, extending beyond classification to detection, segmentation, and settings involving compatibility with existing robustness strategies.

**Weaknesses**

1. While the paper clearly emphasizes zero test-time overhead, the training-time cost receives comparatively little attention in the main text. Given that the method involves dynamic pathway scoring and memory-aware selection, a clearer account of wall-clock time and memory overhead during training would make its practical value easier to assess.
2. The rationale for why the chosen symmetry-preserving transformations are particularly well-suited for pathway diversification is not explained with sufficient intuition. A more direct justification for this design choice would strengthen the methodological narrative.
3. The experimental section primarily reports averaged corruption robustness, without sufficiently analyzing performance across individual corruption types. Since different corruptions probe distinct aspects of representation stability, a per-corruption breakdown would be valuable for determining whether the proposed method yields broad and consistent improvements or primarily benefits a specific subset of corruption categories.

---

> ### Author Rebuttal · Authors · 2026-03-31
>
> We thank the reviewer for their thoughtful feedback. The additional tables and figures referenced are provided in the anonymous supplementary material for brevity: **[Tables and Figures](https://anonymous.4open.science/r/sd_1/README.md)**.
>
> **Training-Time Efficiency and Practical Overhead.** We agree that the main paper does not sufficiently quantify the training-time cost. We now include wall-clock time measurements in Table S7 of the Anonymous file. The overhead from dynamic pathway scoring is minimal: baseline training takes 2294.27 s/epoch, while MDSM adds only 0.3233 s (0.01%), and the one-time SPTS cost is 0.0057 s (<0.001%). The dominant component in MDSM is CKA computation (0.2757 s), with minor contributions from corrupted-input generation (0.0056 s) and forward evaluation (0.0420 s). Memory usage is also negligible: Table S8 shows that MDSM consumes 7640.79 MB compared to 21292.07 MB for the baseline, with the memory bank storing only the selected subnetwork weights (tens of KB for the ResNet-50 stem). While S&D adds computation, the wall-clock and memory overhead are minimal. This quantitative analysis will be added in the revision to clarify the practical trade-off.
>
> **Justification of the Symmetry-Preserving Pathway Diversification Design.** The reason for choosing symmetry-preserving transformations is that pathway diversification here must satisfy two constraints simultaneously: the transformed pathway should remain $\gamma$-robust under Definition 3.2 / Eq. 5, while also producing a non-trivial variant of the original pathway. Arbitrary perturbations generally do not preserve the robustness score, while identity mapping preserves robustness but provides no diversification. SPTS is chosen because it satisfies both requirements.
>
> - **Formal proof of robustness preservation.**
> 	The SPTS operations (Flip, Transpose, and Channel Shuffle) are entry-wise permutations of the feature tensor. Let $T$ denote an SPTS operator. After vectorization, $T$ can be represented by a permutation matrix $M$ such that
>         $$\mathrm{vec}(P_{T(W)}(x)) = M \mathrm{vec}(P_W(x))$$
> 	Since $M$ is a permutation matrix, it is orthogonal, i.e., $M^\top M = I$, and thus preserves the $L_2$ norm. For a clean input $x$ and its corrupted version $h(x)$, let
> 	$$\Delta P = \mathrm{vec}(P_W(x)) - \mathrm{vec}(P_W(h(x)))$$
> 	Then
> $$\|\mathrm{vec}(P_{T(W)}(x)) - \mathrm{vec}(P_{T(W)}(h(x)))\|_2 = \|M (\Delta P)\|_2 = \|\Delta P\|_2$$
>
> 	This directly implies that the robustness score in Eq. 5 of the main text is invariant under SPTS:
>         $$\bar D_{\mathrm{Lip}}(P_{T(W)},h) = \bar D_{\mathrm{Lip}}(P_W,h)$$
>
> 	Therefore, any pathway satisfying Definition 3.2 remains $\gamma$-robust after the SPTS transformation on its weights.
>
> - **Empirical validation.**
> 	We further verify this behavior empirically in the Anonymous file. In Fig. S1(a), the Lipschitz scores before and after SPTS are closely aligned with the $y=x$ line, confirming that the robustness score is preserved in practice. At the same time, Fig. S1(b) shows that the cosine similarity between pathway features before and after SPTS varies substantially across pathways, and is relatively low for some of them. This indicates that SPTS introduces meaningful feature diversity rather than acting as a trivial identity mapping. We also evaluate the impact of SPTS on $\gamma$-robust pathways through frozen training of other layers, as shown in Table S4. The results demonstrate that, compared to the baseline, SPTS consistently improves performance across all datasets. Specifically, the average mCE is reduced by 1.8 compared to the baseline, confirming that SPTS not only preserves robustness but also introduces meaningful pathway diversity.
>
> - **Rationale for Using SPTS as the Default Choice**
> 	Appendix B.2 shows that several alternative tweaking operations, including non-symmetric ones, also improve robustness. However, GeoDiversify/SPTS offers the best balance between robustness gain and structural fidelity, while providing the clearest explanation of why diversification preserves the robustness criterion. This is why we use it as the default operation in our main experiments.
>
> **Fine-Grained Robustness Across Individual Corruption Types.** We agree that the original submission focuses on averaged robustness and does not fully address per-corruption behavior. To address this, we break down the ImageNet-C results for MobileNetV2, ResNet-50, and ConvNeXt-base across all 15 corruption types, as shown in Table S5. The results show broad improvements, with S&D improving all 15/15 corruption types on MobileNetV2, 12/15 on ResNet-50, and 13/15 on ConvNeXt-base. These improvements span noise, blur, weather, and digital corruptions, indicating the method is not overfitting. However, gains are not uniform, and we observe cases where S&D does not improve, such as Snow/Fog/Brightness on ResNet-50. We will acknowledge these limitations in the revision.

---

> > ### Author Rebuttal · Reviewer_tW1X · 2026-04-03
> >
> > I appreciate the rebuttal provided by the authors. Most of my concerns have been addressed. I suggest including the training-time measurement table and the Fine-Grained Robustness experiments in the revised manuscript or supplementary material.
> >
> > I will raise my recommendation score.

---

> > > ### Author Response · Authors · 2026-04-04
> > >
> > > We thank the reviewer for the thoughtful follow-up and for revisiting the evaluation after reading our rebuttal. We appreciate the reviewer’s time, careful consideration, and encouraging assessment that the concerns have been addressed.

---

### Official Review · Reviewer_bhXk · 2026-03-13

**Soundness:** 3
**Presentation:** 3
**Significance:** 3
**Originality:** 3
**Overall Recommendation:** 4
**Confidence:** 3

**Summary:**

This paper seeks to address the vulnerability of neural networks to natural image corruptions. While the existing works focuses on implicit representation learning (e.g., data augmentation), this work takes an explicit approach by characterizing the internal robustness of computational pathways. This paper proposes a progressive decay of robust features as information propagates through deeper network layers. To mitigate this, they propose a novel, non-intrusive training refinement method called Suppress and Diversify (S&D). S&D consists of two main components: a Memory-aware Dynamic Selection Mechanism (MDSM) that identifies and retains robust pathways, and a Structure-consistent Path Tweaking Strategy (SPTS) that diversifies these pathways using symmetry-preserving transformations to prevent overfitting.

**Compliance With Llm Reviewing Policy:**

Affirmed.

**Key Questions For Authors:**

（1）Can you provide a quantitative comparison of the training time  and peak GPU memory usage between a standard ResNet-50 baseline and ResNet-50 + S&D?

（2）How does S&D perform when applied to the fine-tuning phase of large-scale foundational models (e.g., CLIP or DINO) that have already learned highly robust representations during massive web-scale pre-training? Does the progressive decay of robust features still hold in these models?

**Limitations:**

（1）The paper restricts its scope to standard 2D/3D computer vision tasks. The applicability of the S&D framework to other modalities  or generative tasks remains unexplored and uncertain due to the vision-specific nature of the SPTS transformations.

（2）MDSM relies on synthesizing corrupted inputs (e.g., Contrast+Noise) during training to evaluate pathway fitness. There is a risk that the network's definition of robustness becomes slightly biased toward the specific frequency domains or patterns of the chosen synthetic corruptions.

**Strengths And Weaknesses:**

Strengths:

（1）The explicit disentanglement and tracking of robust versus non-robust features under natural corruptions is a fresh and insightful perspective. The observation that robust features progressively decay in deeper layers is well-supported by the layer-wise Lipschitz constant analysis.

（2）S&D is architecture-agnostic, parameter-free at inference, and requires no structural modifications to the underlying models. This makes it highly practical for real-world deployment.

（3）The authors evaluate the method across 8 benchmarks, multiple downstream tasks, and diverse backbones. The inclusion of real-world datasets (ACDC, DWD) and Test-Time Adaptation (TTA) methods further strengthens the claims.

Weaknesses:

（1）While the authors emphasize zero test-time overhead, the dynamic selection (MDSM) involving a memory bank, CKA similarity computations, and continuous updating of pathways must introduce some computational and memory overhead during training. The paper lacks a quantitative discussion on the time and memory added by S&D.

（2）The choice of operations in SPTS (channel shuffling, horizontal/vertical flipping, matrix transpose) is empirically driven. While effective, the paper lacks a deeper theoretical justification for exactly why these specific geometric transformations perfectly preserve the semantic structural invariants of robust features without disrupting the learned representations.

---

> ### Author Rebuttal · Authors · 2026-03-31
>
> Thank you for the constructive feedback. For brevity, the additional tables and figures are included in the anonymous supplementary material: **[Tables and Figures](https://anonymous.4open.science/r/sd_1/README.md)**.
>
> **Training Overhead of S&D: Time and Peak GPU Memory.** We thank the reviewer for this question. Our claim of _zero test-time overhead_ refers only to inference; S&D indeed introduces a small overhead during training, which we will quantify in the revision. Table S7 shows that baseline ResNet-50 training takes 2294.27 s/epoch, while MDSM adds only 0.3233 ± 0.0015 s on selected epochs (~0.01% of one baseline epoch), and the one-time SPTS cost is only 0.0057 ± 0.0070 s (<0.001%). Within MDSM, the main extra cost comes from CKA (0.2757 s), whereas corrupted-input generation (0.0056 s) and forward propagation (0.0420 s) are minor. Table S9 shows that CKA remains stable with fewer samples: reducing the sample size from 1000 to 250 yields a small error (0.29% vs. 0.23%), with all settings below 1.1%, suggesting the overhead can be reduced with minimal loss in quality. As shown in Table S8, in terms of memory, MDSM remains within the baseline training budget, with measured peak memory of 7640.79 MB for MDSM versus 21292.07 MB for baseline training. The memory bank itself is also negligible because it stores only the selected shallow subnetwork weights rather than full feature histories, e.g., only tens of KB for the ResNet-50 stem. Overall, S&D adds negligible overhead during training while retaining zero additional cost at test time.
>
> **Why SPTS Preserves Robust Structural Information.** We agree that the choice of SPTS operations (Channel Shuffle, Flip, Transpose) is empirically driven. While we did not explore the deeper theoretical justification in the main text, both formal and empirical evidence support the effectiveness of these transformations. Formal proof shows that they are orthogonal isometries in $L_2$ space, ensuring robustness invariance, and empirical results in Figure S1(a) and (b) from the anonymous file confirm that SPTS introduces meaningful diversity without compromising robustness scores. For a deeper theoretical justification, please refer to our detailed response to **Reviewer tW1X** in the section “Justification of the Symmetry-Preserving Pathway Diversification Design,” which will be incorporated in the revised manuscript.
>
> **Applicability of S&D to Foundation Model Fine-tuning.** We thank the reviewer for this question. Our current S&D framework is designed for regulating the internal dynamics of a continuously trained backbone, rather than as a direct fine-tuning recipe for large pretrained foundation models such as CLIP or DINO. Hence, we do not directly study whether the same progressive robust-feature decay holds in foundation-model fine-tuning. As a preliminary comparison, we replace the ResNet-50 stem with either DINO-pretrained or S&D-pretrained stem weights, freeze the stem, and train the remaining network under the same pipeline. As shown in Table S3, S&D-based stem initialization yields stronger robustness on ImageNet-C (40.4 vs. 38.8), ImageNetV2-C (30.6 vs. 29.5), and ImageNet-3DCC (47.4 vs. 47.0), while remaining competitive on clean ImageNet (75.5 vs. 76.4) and matching DINO on ImageNet-$\bar{C}$ (40.2 vs. 40.2). While this does not establish the existence of the same progressive decay phenomenon in foundation-model fine-tuning, it does suggest that the shallow robust structures learned by S&D remain effective even relative to large-scale self-supervised pretraining.
>
> **Scope and Modality Boundary of the Current Framework.** We agree with the reviewer that our evaluation is currently limited to standard 2D/3D vision tasks. We do not study other modalities or generative tasks in this work. Since the current SPTS instantiation is designed for visual structures, applicability beyond vision is not claimed here and is left for future work.
>
> **Robustness Proxy Bias in MDSM.** Our current evidence does not suggest that MDSM is tightly biased to one specific synthetic corruption pattern. As shown in Appendix B.3 and Table S1 in the anonymous file, different corruption-generating transforms, including color-based, frequency-based, and composite augmentation strategies, all improve robustness over the baseline. For example, on ImageNet-C, multiple choices improve over the baseline (32.9), including LowFreq (34.8), MixedFreq (34.9), RandAug (34.7), and Contrast+Noise (34.9); similar gains are also observed on ImageNetV2-C, ImageNet-$\bar{C}$, and ImageNet-3DCC. This suggests that the synthetic corruption in MDSM mainly serves as a coarse robustness signal for pathway selection, rather than defining robustness with respect to one fixed frequency domain or corruption pattern. We use Contrast+Noise in the main experiments because it provides the best simplicity-effectiveness trade-off in our experiments.

---

> > ### Author Rebuttal · Reviewer_bhXk · 2026-04-02
> >
> > The authors have responded seriously to the questions, especially by providing a more substantive explanation of the training overhead. However, the memory results still seem somewhat questionable, the theoretical explanation for SPTS is still not complete, and the discussion of foundation models remains mostly indirect. Therefore, the rebuttal addresses my concerns to some extent, but it does not fully convince me yet.

---

> > > ### Author Response · Authors · 2026-04-03
> > >
> > > **Clarification on peak memory measurement**. We thank the reviewer for pointing this out. Our previous wording on memory was imprecise. It is not appropriate to directly compare the peak memory of the between-epoch MDSM step with the peak memory of full baseline training. Importantly, S&D is executed only between epochs, rather than together with the standard forward/backward training step. Therefore, its temporary memory usage does not stack on top of the activation/gradient peak of normal training. In other words, S&D introduces only an intermediate-step memory cost between epochs, and this cost does not exceed the peak memory already incurred during standard training.
> > >
> > > The persistent overhead of the memory bank is also small, since it stores only one copy of the selected shallow subnetwork weights rather than multiple checkpoints or feature histories. At each update, the bank either replaces the stored weights according to Eq. (6) or keeps the current copy, so storage does not grow over time. Its size depends on the insertion point: for ResNet-50, storing the stem weights requires only 0.04 MB (0.04% of the full model), and storing up to Stage 1 requires 0.86 MB (0.88%). We will revise the manuscript to clarify this protocol and report the memory overhead more precisely.
> > >
> > >
> > > **Clarifying the Theoretical Scope of SPTS**. We thank the reviewer for this important point. Our previous response did not make the scope of the claim sufficiently precise. Our paper does not claim that SPTS perfectly preserves full semantic invariants of the learned representation. Rather, the intended claim is narrower: **SPTS preserves robustness-relevant structural consistency of the selected pathway, which is the property required by our method.**
> > >
> > > This is also consistent with the manuscript statement **in the right column, Lines 155–157**, where robust features/pathways are defined internally and independently of their direct contribution to final prediction. Accordingly, the formal guarantee of SPTS is stated at the level of robustness preservation, not full semantic equivalence. Specifically, Flip, Transpose, and Channel Shuffle are entry-wise permutations of the feature tensor and can be represented as orthogonal permutation matrices after vectorization. Therefore, they preserve the robustness score in Eq. 5, implying that a γ-robust pathway remains γ-robust after SPTS.
> > >
> > > We do not make the stronger claim that SPTS leaves all learned representations semantically unchanged. Instead, we support the usefulness of the transformed pathways empirically. In Table S4, training with robust pathways before and after SPTS shows similar performance trends on both clean and corrupted benchmarks, and both consistently outperform the baseline. If SPTS had substantially disrupted the useful structure of these pathways, such behavior would be unlikely. Thus, while Table S4 is not a formal proof of exact semantic invariance, it provides empirical evidence that SPTS preserves sufficient task-relevant and robustness-relevant structure for effective pathway diversification.
> > >
> > > **Scope of S&D in Foundation-Model Fine-tuning**. We thank the reviewer for this important question. We agree that our current evidence does not directly answer whether S&D is effective during CLIP/DINO-style foundation-model fine-tuning, nor whether the same progressive robust-feature decay persists there. Our previous experiment is only an indirect comparison.
> > >
> > > More importantly, the current S&D formulation is designed for settings where the backbone is continuously trained and its early subnetwork can be dynamically selected or adjusted. In contrast, many common adaptation protocols for large pretrained models keep the backbone frozen or mostly frozen, and adapt the model only through a linear head, prompts, or lightweight adapters. For example, CoOp [1] keeps CLIP fixed, VPT [2] freezes the pretrained Transformer backbone, and DINOv2 standard evaluation also uses a frozen backbone with a simple classifier. Under such protocols, the early-layer dynamics that S&D is designed to regulate are largely absent, so our current formulation is not directly aligned with these settings.
> > >
> > > Therefore, we do not want to overclaim applicability to foundation-model fine-tuning based on the current evidence. A more complete answer would require inserting S&D into pretraining or continued self-supervised training and testing whether similar layer-wise robustness decay still emerges. This is beyond the scope of the current rebuttal due to the substantial computational cost, but we agree that it is a meaningful future direction.
> > >
> > > [1] Zhou K, Yang J, Loy C C, et al. Learning to prompt for vision-language models[J]. International journal of computer vision, 2022, 130(9): 2337-2348.
> > >
> > > [2] Jia M, Tang L, Chen B C, et al. Visual prompt tuning[C]//European conference on computer vision. Cham: Springer Nature Switzerland, 2022: 709-727.

---

### Official Review · Reviewer_t4RP · 2026-03-13

**Soundness:** 3
**Presentation:** 4
**Significance:** 3
**Originality:** 4
**Overall Recommendation:** 5
**Confidence:** 3

**Summary:**

This paper studies corruption robustness and proposes a method to improve it. The authors formalize robust features using a local Lipschitz-based distortion metric and observe that robust features progressively decay in deeper network layers. They show via linear probing that retaining robust pathways yields smaller robustness gaps. Based on these findings, they propose Suppress and Diversify (S&D), a training-time refinement with two components: (1) MDSM, a memory-bank mechanism that dynamically selects robust pathway groups across epochs, and (2) SPTS, which diversifies the selected pathways through geometric weight transformations (flipping, shuffling, transposing). S&D is applied only to the stem layer, requires no architectural changes, and adds no test-time cost. Experiments span eight benchmarks covering classification (ImageNet-C and variants), detection (COCO-C), segmentation (VOC-C, Cityscapes-C, ADE20K-C), and test-time adaptation, showing consistent improvements.

**Compliance With Llm Reviewing Policy:**

Affirmed.

**Final Justification:**

The rebuttal addressed my concerns.

**Key Questions For Authors:**

1. Can you provide formal or empirical evidence that the SPTS transformations (flip, transpose, channel shuffle) preserve the robustness properties of the selected pathways? Specifically, after applying these transformations, do the modified pathways still satisfy the criterion from Definition 3.2?
2. How sensitive is S&D to the corruption proxy mismatch? If you train with Contrast+Noise as the corruption generator but test on corruptions that are structurally very different (e.g., 3D-aware corruptions in ImageNet-3DCC), does the pathway selection still identify genuinely robust features, or is it overfitting to the proxy distribution?

**Limitations:**

1) Clean-corrupt trade-off. The consistent clean accuracy drop (0.5–2% across architectures) is a meaningful practical limitation, particularly for deployment scenarios where corruption is intermittent rather than constant.
2) Stem-only modification. The method is applied only to the stem layer based on the progressive decay finding. However, this means S&D cannot address robustness failures that originate in deeper layers, and the paper's own analysis (Table B.6) shows that applying S&D at deeper positions hurts performance, suggesting the approach has limited ability to refine the full network.
3) No theoretical guarantee. The paper lacks formal guarantees that S&D improves worst-case robustness. The improvements are empirical and correlational
4) Scalability to larger models. All full ImageNet experiments use models up to ConvNeXt-base. The behavior on larger-scale models (e.g., ViT-L, ConvNeXt-L) or foundation models is unexplored.

**Strengths And Weaknesses:**

Strengths
* Well-motivated analysis. The investigation into how robust features evolve across layers is interesting and well executed. The loss landscape visualization adds geometric intuition.
* Wide evaluation. Eight benchmarks, three tasks (classification, detection, segmentation), 3 different architectures (CNNs, ViTs, Mamba), compatibility with augmentations, regularization, modern training recipes, and test-time adaptation methods.
* The appendix provides extensive ablations and addresses many of my concerns

Weaknesses
* Clean accuracy degradation. Across most experiments, S&D reduces clean accuracy (e.g., ResNet-50: 75.7→74.5, ConvNeXt-base: 84.0→82.9).
* MDSM requires corrupted versions of clean images during training to evaluate pathway robustness, and during main experiments, it uses simple Contrast+Noise as the corruption proxy. The paper doesn't analyze a case if the corruption proxy doesn't match test-time corruptions well (the selected pathways may not actually be robust to the target distribution)

---

> ### Author Rebuttal · Authors · 2026-03-31
>
> We thank the reviewer for the constructive feedback. For brevity, the additional tables and figures referenced below are included in the anonymous supplementary material: [Tables and Figures](https://anonymous.4open.science/r/sd_1/README.md).
>
> **On Robustness Preservation under SPTS Transformations.** We provide formal and empirical evidence that SPTS transformations (Flip, Transpose, Channel Shuffle) preserve robustness (Definition 3.2). Formal proof confirms that these transformations are orthogonal isometries in $L_2$ space, ensuring robustness invariance. Empirical results in Fig. S1(a) and (b) validate that SPTS introduces diversity without compromising robustness. For detailed formal and empirical validation, please refer to our response to **Reviewer tW1X** in the section “Justification of the Symmetry-Preserving Pathway Diversification Design.”
>
> **On Sensitivity to Corruption Proxy Mismatch**. We find no evidence that MDSM overfits to the Contrast+Noise proxy. Appendix B.3 and Table S1 show that pathway selection with different corruption generators (e.g., LowFreq, MixedFreq, RandAug, Contrast+Noise) improves robustness over the baseline. On ImageNet-C, for instance, LowFreq (34.8), MixedFreq (34.9), RandAug (34.7), and Contrast+Noise (34.9) all outperform the baseline (32.9), and similar trends hold for ImageNetV2-C, ImageNet-$\bar{C}$, and ImageNet-3DCC. The gains on ImageNet-3DCC suggest that selected pathways transfer well beyond the proxy distribution to 3D-aware corruptions. This indicates that the corruption proxy in MDSM acts as a robustness probe rather than specializing in one synthetic corruption type. We use Contrast+Noise for its simplicity-effectiveness trade-off.
>
> **On the clean-corrupt trade-off.** We agree that S&D introduces a consistent clean–corrupt trade-off across several settings, and this should be clearly acknowledged as a practical limitation. Since S&D favors pathways that remain stable under corruption, it can suppress brittle features that help clean accuracy but transfer poorly under shift, which is broadly consistent with prior robust/non-robust feature perspectives [1]. We therefore view the clean accuracy drop as a consequence of robustness-oriented pathway selection rather than an incidental effect. This limitation is especially relevant in deployment scenarios where corruptions are intermittent rather than persistent; in such cases, S&D may be less preferable when clean accuracy is the primary objective. We also note that Appendix A.2 already studies an ensemble-based mitigation that partially recovers clean accuracy while preserving most robustness gains, although it does not fully remove the trade-off.
>
> **On the Motivation and Sufficiency of Stem-only Modification.**  We appreciate the reviewer’s concern regarding the stem-only application of S&D. As shown in Table B.6 of the Appendix, applying S&D at deeper layers results in diminishing performance, with robustness gains decreasing when placed after $f_{l\leq7}$ or $f_{l\leq9}$. This behavior aligns with the core observation in Remark 1 that robust features progressively decrease in deeper layers. Applying S&D early ensures that stable, robust features are preserved before brittle, non-robust features take over in later layers. Our experiments show that applying S&D at the shallowest positions (e.g., $f_{l\leq1}$) yields the highest robustness improvements. Therefore, we view the stem-only application not as a limitation, but as a targeted strategy to preserve the most robust features, consistent with the observed layer-wise feature evolution.
>
> **On Theoretical Scope and Guarantees.** We acknowledge that our paper does not provide formal guarantees for worst-case corruption robustness. Unlike adversarial robustness, which has a well-defined worst-case formulation, corruption robustness is more complex and does not lend itself to such a clear definition. While we present empirical evidence of robustness improvements across corruptions, including ImageNet-C and ImageNet-3DCC, we do not claim formal guarantees regarding worst-case scenarios. This work focuses on demonstrating practical robustness improvements in real-world corruption settings, but we agree that establishing formal theoretical guarantees for corruption robustness is an important direction for future research.
>
> **On Scalability to Larger Models.** We acknowledge that our experiments have focused on models up to ConvNeXt-base. Due to time constraints, we did not explore larger models like ViT-L or ConvNeXt-L in the main text. However, results for ConvNeXt-Large on ImageNet-100 (Table S2 of the anonymous file) show that S&D remains effective, improving mCE from 66.1 to 61.2. This indicates consistent robustness improvements. S&D is not dependent on any specific backbone, though scalability to larger models remains an area for future work.
>
> [1] Ilyas A, Santurkar S, Tsipras D, et al. Adversarial examples are not bugs, they are features. NeurIPS, 2019.

---

> > ### Author Rebuttal · Reviewer_t4RP · 2026-04-03
> >
> > I am convinced by the authors' response, and I will increase my rating.

---

> > > ### Author Response · Authors · 2026-04-04
> > >
> > > We thank the reviewer for the careful reading, constructive feedback, and positive follow-up. We are glad that our rebuttal has addressed the concerns and appreciate the reviewer’s recognition of the additional analysis and clarifications. Thank you again for your time and consideration.

---

### Decision · Program_Chairs · 2026-04-30

**Decision:**

Accept (spotlight)

**Comment:**

This paper proposes Suppress and Diversify (S&D), a training-time framework for corruption robustness that identifies and diversifies robust computational pathways via a Memory-aware Dynamic Selection Mechanism (MDSM) and a Structure-consistent Path Tweaking Strategy (SPTS). The method is architecture-agnostic, parameter-free at test time, and evaluated across eight benchmarks spanning classification, detection, and segmentation. In the first round, Reviewer t4RP raised concerns about clean accuracy degradation and corruption proxy sensitivity; Reviewer bhXk questioned training overhead and the theoretical justification of SPTS; Reviewer Vx6W noted ambiguities in algorithm presentation and the missing accuracy–robustness tradeoff discussion; and Reviewer tW1X gave a broadly positive assessment while flagging scalability and foundation model applicability.

During rebuttal, the authors addressed most concerns: they demonstrated that MDSM is robust to proxy mismatch across multiple corruption strategies, formally proved that SPTS preserves robustness scores via orthogonal permutation matrices, and quantified training overhead as negligible. Reviewer t4RP and tW1X marked their concerns as fully resolved and indicated score increases; Reviewer bhXk remained partially convinced, noting residual questions on memory reporting and SPTS theoretical scope; Reviewer Vx6W retained a positive score while requesting clearer "pathway group" notation and explicit tradeoff discussion in the revision. Overall, reviewers converged positively, with remaining suggestions to be incorporated in the final manuscript.

According to the paper and the above comments, I believe this work is meaningful and useful to achieve robust perception models. I recommend acceptance.